# Graph Convolutional Normalizing Flows for Semi-Supervised Classification & Clustering

## Abstract

Graph neural networks (GNNs) are *discriminative models* that directly model the class posterior $p(y|\mathbf{x})$ for semi-supervised classification of graph data. While being effective for prediction, as a representation learning approach, the node representations extracted from a GNN often miss useful information for effective clustering, because that is not necessary for a good classification. In this work, we replace a GNN layer by a combination of graph convolutions and normalizing flows under a Gaussian mixture representation space, which allows us to build a *generative model* that models both the class conditional likelihood $p(\mathbf{x}|y)$ and the class prior $p(y)$. The resulting neural network, GC-Flow, enjoys two benefits: it not only maintains the predictive power because of the retention of graph convolutions, but also produces well-separated clusters in the representation space, due to the structuring of the representation as a mixture of Gaussians. We demonstrate these benefits on a variety of benchmark data sets. Moreover, we show that additional parameterization, such as that on the adjacency matrix used for graph convolutions, yields additional improvement in clustering.

## 1 Introduction

Semi-supervised learning (Zhu, 2008) refers to the learning of a classification model by using typically a small amount of labeled data with possibly a large amount of unlabeled data. The presence of the unlabeled data, together with additional assumptions (such as the manifold and smoothness assumptions), may significantly improve the accuracy of a classifier learned even with few labeled data. A typical example of such a model in the recent literature is the graph convolutional network (GCN) of Kipf & Welling (2017), which capitalizes on the graph structure (considered as an extension of a discretized manifold) underlying data to achieve effective classification. GCN, together with other pioneering work on parameterized models, have formed a flourishing literature of graph neural networks (GNNs), which excel at node classification (Zhou et al., 2020; Wu et al., 2021).

However, driven by the classification task, GCN and other GNNs may not produce node representations with useful information for goals different from classification. For example, the representations do not cluster well in some cases. Such a phenomenon is of no surprise. For instance, when one treats the penultimate activations as the data representations and uses the last dense layer as a linear classifier, the representations need only be close to linearly separable for an accurate classification; they do not necessarily form well-separated clusters.

This observation leads to a natural question: can one build a representation model for graphs that not only is effective for classification but also unravels the inherent structure of data for clustering? The answer is affirmative. One idea is to, rather than construct a discriminative model $p(y|\mathbf{x})$ as all GNNs do, build a generative model $p(\mathbf{x}|y)p(y)$ whose class conditional likelihood is defined by explicitly modeling the representation space, for example by using a mixture of well-separated unimodal distributions. Indeed, the recently proposed FlowGMM model (Izmailov et al., 2020) uses a normalizing flow to map the distribution of input features to a Gaussian mixture, resulting in well-structured clusters. This model, however, is not designed for graphs and it underperforms GNNs that leverage the graph structure for classification.

In this work, we present *graph convolutional normalizing flows* (GC-Flows), a generative model that not only classifies well, but also yields node representations that capture the inherent structure of data, as a result forming high-quality clusters. We can relate GC-Flows to both GCNs and

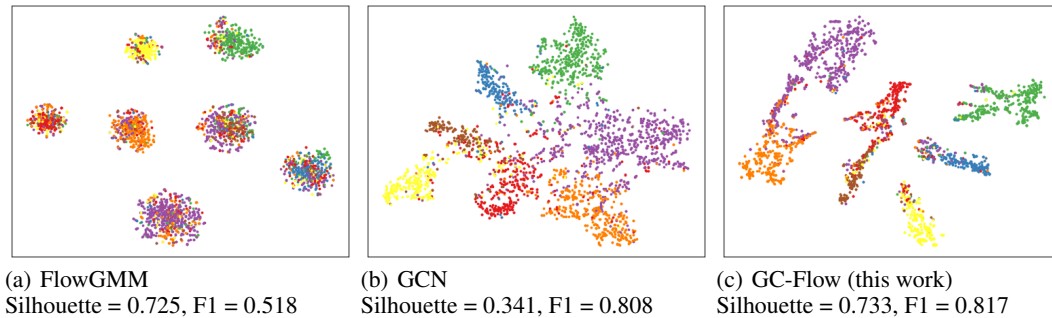

(a) FlowGMM
Silhouette = 0.725, F1 = 0.518

(b) GCN
Silhouette = 0.341, F1 = 0.808

(c) GC-Flow (this work)
Silhouette = 0.733, F1 = 0.817

Figure 1: Representation space of the data set Cora under different models, visualized by t-SNE. Coloring indicates groud-truth labeling. Silhouette coefficients measure cluster separation. Micro-F1 scores measure classification accuracy.

FlowGMMs. On the one hand, GC-Flows incorporate each GCN layer with an invertible flow. Such a flow parameterization allows training a model through maximizing the likelihood of data representations being a Gaussian mixture, mitigating the poor clustering effect of GCNs. On the other hand, GC-Flows augment a usual normalizing flow model (such as FlowGMM) that is trained on independent data, with one that incorporates graph convolutions as an inductive bias in the parameterization, boosting the classification accuracy. In Figure 1, we visualize for a graph data set the nodes in the representation space using t-SNE. It suggests that GC-Flow inherits the clustering effect of FlowGMM, while being similarly accurate to GCN for classification.

A few key characteristics of GC-Flows are as follows:

1. A GC-Flow is a GNN, because being applied to graph data, it computes node representations by using the graph structure. In contrast, a FlowGMM is not a GNN.

2. A GC-Flow is a generative model, admitting FlowGMMs as a special case when graph is absent.

3. As a generative model, the training loss function of GC-Flows involves both labeled and unlabeled data, similar to FlowGMMs, while that of GNNs involves only the labeled data.

**Significance.** While classification is the dominant node-level task that concerns the current literature on GNNs, the importance of clustering in capturing the inherent structure of data is undeniable. This work addresses a weakness of the current GNN literature—particularly, the separation of clusters. A Gaussian mixture representation space properly reflects this goal. The normalizing flow is a vehicle to parameterize the feature transformation, so that it encourages the formation of separated Gaussians. It is a generative model that can return data densities. The likelihood training is organically tied to a generative model, whereas existing methods based on clustering or contrastive losses externally encourage the GNN to produce clustered representations, without a notation of densities.

## 2 RELATED WORK

Graph neural networks (GNNs) are machineries to produce node-level and graph-level representations, given graph-structured data as input (Zhou et al., 2020; Wu et al., 2021). A popular class of GNNs are message passing neural networks (MPNNs) (Gilmer et al., 2017), which treat information from the neighborhood of a node as messages and recursively update the node representation through aggregating the neighborhood messages and combing the result with the past node representation. Many popularly used GNNs can be considered a form of MPNNs, such as GG-NN (Li et al., 2016), GCN (Kipf & Welling, 2017), GraphSAGE (Hamilton et al., 2017), GAT (Veličković et al., 2018), and GIN (Xu et al., 2019).

Normalizing flows are invertible neural networks that can transform a data distribution to a typically simple one, such as the normal distribution (Rezende & Mohamed, 2015; Kobyzev et al., 2021; Papamakarios et al., 2021). Because of invertibility, one may navigate the input and output distributions for purposes such as estimating densities and sampling new data. The densities of the two distributions are related by the change-of-variable formula, which involves the Jacobian determinant of the flow. Computing the Jacobian determinant is costly in general; thus, many proposed neural

networks exploit constrained structures, such as the triangular pattern of the Jacobian, to reduce the computational cost. Notable examples include NICE (Dinh et al., 2015), IAF (Kingma et al., 2016), MAF (Papamakarios et al., 2017), RealNVP (Dinh et al., 2017), Glow (Kingma & Dhariwal, 2018), and NSF (Durkan et al., 2019). While these network mappings are composed of discrete steps, another class of normalizing flows with continuous mappings have also been developed, which use parameterized versions of differential equations (Chen et al., 2018b; Grathwohl et al., 2019).

Normalizing flows can be used for processing or creating graph-structured data in different ways. For example, GraphNVP (Madhawa et al., 2019) and GraphAF (Shi et al., 2020) are graph generative models that use normalizing flows to generate a graph and its node features in a one-shot and a sequential manner, respectively. GANF (Dai & Chen, 2022) uses an acyclic directed graph to factorize the otherwise intractable joint distribution of time series data and uses the estimated data density to detect anomalies. GNF (Liu et al., 2019) is both a graph generative model and a graph neural network. For the latter functionality, GNF is relevant to our model, but its purpose is to classify rather than to cluster, thus missing the representation-space modeling and a training objective suitable for clustering. Furthermore, the architecture of GNF differs from ours in the role the graph plays. For our method, the graph adjacency matrix is part of the flow mapping, hence provoking a determinant calculation with respect to the matrix; whereas for GNF, the graph is used in the parameterization of an affine coupling layer and it incurs no determinant calculation. CGF (Deng et al., 2019) extends the continuous version of normalizing flows to graphs, where the dynamics of the differential equation is parameterized as a message passing layer. The difference between our model and CGF inherits the general difference between discrete and continuous flows in how different parameterizations transform distributions.

Under the focus on clustering, several graph-based methods were developed based on the use of GNNs for feature extraction. For example, Fettal et al. (2022) use a combination of reconstruction and clustering losses to train the GNN; whereas Zhu et al. (2021); Li et al. (2022); Jing et al. (2022) use contrastive losses. Different from ours, these methods do not model the data (or representation) space with distributions as generative methods do. We empirically compare with several contrastive methods and demonstrate that our model significantly outperforms them in cluster separation.

## 3 Preliminaries

In this section, we review a few key concepts and familiarize the reader with notations to be used throughout the paper.

### 3.1 Normalizing Flow

Let $\mathbf{x} \in \mathbb{R}^D$ be a $D$-dimensional random variable. A *normalizing flow* is a vector-valued invertible mapping $\mathbf{f}(\mathbf{x}) : \mathbb{R}^D \to \mathbb{R}^D$ that normalizes the distribution of $\mathbf{x}$ to some base distribution, whose density is easy to evaluate. Let such a base distribution have density $\pi(\mathbf{z})$, where $\mathbf{z} = \mathbf{f}(\mathbf{x})$. With the change-of-variable formula, the density of $\mathbf{x}$, $p(\mathbf{x})$, can be computed as

$$p(\mathbf{x}) = \pi(\mathbf{f}(\mathbf{x}))|\det \nabla \mathbf{f}(\mathbf{x})|, \tag{1}$$

where $\nabla \mathbf{f}$ denotes the Jacobian of $\mathbf{f}$. In general, such a flow $\mathbf{f}$ may be the composition of $T$ constituent flows, all of which are invertible. In notation, we write $\mathbf{f} = \mathbf{f}_T \circ \mathbf{f}_{T-1} \circ \cdots \circ \mathbf{f}_1$, where $\mathbf{f}_i(\mathbf{x}^{(i-1)}) = \mathbf{x}^{(i)}$ for all $i$, and $\mathbf{x}^{(0)} \equiv \mathbf{x}$ and $\mathbf{x}^{(T)} \equiv \mathbf{z}$. Then, the chain rule expresses the Jacobian determinant as a product of the Jacobian determinants of each constituent flow: $\det \nabla \mathbf{f}(\mathbf{x}) = \prod_{i=1}^{T} \det \nabla \mathbf{f}_i(\mathbf{x}^{(i-1)})$.

In practical uses, the Jacobian determinant of each constituent flow needs be easy to compute, so that the density $p(\mathbf{x})$ in (1) can be evaluated. One example that serves such a purpose is the *affine coupling layer* of Dinh et al. (2017). For notational simplicity, we denote such a coupling layer by $\mathbf{g}(\mathbf{x}) = \mathbf{y}$, which in effect computes

$$\mathbf{y}_{1:d} = \mathbf{x}_{1:d}, \qquad \mathbf{y}_{d+1:D} = \mathbf{x}_{d+1:D} \odot \exp(\mathbf{s}(\mathbf{x}_{1:d})) + \mathbf{t}(\mathbf{x}_{1:d}),$$

where $d = \lfloor D/2 \rfloor$ and $\mathbf{s}, \mathbf{t} : \mathbb{R}^d \to \mathbb{R}^{D-d}$ are any neural networks. It is simple to see that the Jacobian is a triangular matrix, whose diagonal has value $1$ in the first $d$ entries and $\exp(\mathbf{s})$ in the remaining $D - d$ entries. Hence, the Jacobian determinant is simply the product of the exponential of the outputs of the $\mathbf{s}$-network; that is, $\det \nabla \mathbf{g}(\mathbf{x}) = \prod_{i=1}^{D-d} \exp(s_i)$.

## 3.2 GAUSSIAN MIXTURE AND FLOWGMM

Different from a majority of work that take the base distribution in a normalizing flow to be a single Gaussian, we consider it to be a Gaussian mixture, because this is a natural probabilistic model for clustering. Using $k$ to index mixture components ($K$ in total), we express the base density $\pi(\mathbf{z})$ as

$$\pi(\mathbf{z}) = \sum_{k=1}^{K} \phi_k \mathcal{N}(\mathbf{z}; \boldsymbol{\mu}_k, \boldsymbol{\Sigma}_k) \quad \text{with} \quad \mathcal{N}(\mathbf{z}; \boldsymbol{\mu}_k, \boldsymbol{\Sigma}_k) = \frac{\exp(-\frac{1}{2}(\mathbf{z} - \boldsymbol{\mu}_k)^T \boldsymbol{\Sigma}_k^{-1}(\mathbf{z} - \boldsymbol{\mu}_k))}{(2\pi)^{D/2}(\det \boldsymbol{\Sigma}_k)^{1/2}}, \quad (2)$$

where $\phi_k \geq 0$ are mixture weights that sum to unity and $\boldsymbol{\mu}_k$ and $\boldsymbol{\Sigma}_k$ are the mean vector and the covariance matrix of the $k$-th component, respectively.

A broad class of semi-supervised learning models specifies a generative process for each data point $\mathbf{x}$ through defining $p(\mathbf{x}|y)p(y)$, where $p(y)$ is the prior class distribution and $p(\mathbf{x}|y)$ is the class conditional likelihood for data. Then, by the Bayes' Theorem, the class prediction model $p(y|\mathbf{x})$ is proportional to $p(\mathbf{x}|y)p(y)$. Among them, FlowGMM (Izmailov et al., 2020) makes use of the flow transform $\mathbf{z} = \mathbf{f}(\mathbf{x})$ and defines $p(\mathbf{x}|y = k) = \mathcal{N}(\mathbf{f}(\mathbf{x}); \boldsymbol{\mu}_k, \boldsymbol{\Sigma}_k)|\det \nabla \mathbf{f}(\mathbf{x})|$ with $p(y = k) = \phi_k$. This definition is valid, because marginalizing over the class variable $y$, one may verify that $p(\mathbf{x}) = \sum_y p(\mathbf{x}|y)p(y)$ is consistent with the density formula (1), when the base distribution follows (2).

## 3.3 GRAPH CONVOLUTIONAL NETWORK

The GCNs (Kipf & Welling, 2017) are a class of parameterized neural network models that specify the probability of class $y$ of a node $\mathbf{x}$, $p(y|\mathbf{x})$, collectively for all nodes $\mathbf{x}$ in a graph, without defining the data generation process as in FlowGMM. To this end, we let $\mathbf{A} \in \mathbb{R}^{n \times n}$ be the adjacency matrix of the graph, which has $n$ nodes, and let $\mathbf{X} = [\mathbf{x}_1, \cdots, \mathbf{x}_n]^T \in \mathbb{R}^{n \times D}$ be the input feature matrix, with $\mathbf{x}_i$ being the feature vector for the $i$-th node. We further let $\mathbf{P} \in \mathbb{R}^{n \times K}$ be the output probability matrix, where $K$ is the number of classes and $\mathbf{P}_{ik} \equiv p(y = k|\mathbf{x}_i)$. An $L$-layer GCN is written as

$$\mathbf{X}^{(i)} = \sigma_i(\widehat{\mathbf{A}} \mathbf{X}^{(i-1)} \mathbf{W}^{(i-1)}), \quad i = 1, \ldots, L, \quad (3)$$

where $\mathbf{X} \equiv \mathbf{X}^{(0)}$ and $\mathbf{P} \equiv \mathbf{X}^{(L)}$. Here, $\sigma_i$ is an element-wise activation function, such as ReLU, for the intermediate layers $i < L$, while $\sigma_L$ is the row-wise softmax activation function for the final layer. The matrices $\mathbf{W}^{(i)}$, $i = 0, \ldots, L - 1$, are learnable parameters and $\widehat{\mathbf{A}}$ denotes a certain normalized version of the adjacency matrix $\mathbf{A}$. The standard definition of $\widehat{\mathbf{A}}$ for an undirected graph is $\widehat{\mathbf{A}} = \widetilde{\mathbf{D}}^{-\frac{1}{2}} \widetilde{\mathbf{A}} \widetilde{\mathbf{D}}^{-\frac{1}{2}}$, where $\widetilde{\mathbf{A}} = \mathbf{A} + \mathbf{I}$ and $\widetilde{\mathbf{D}} = \text{diag}\left(\sum_j \widetilde{\mathbf{A}}_{ij}\right)$, but we note that many other variants of $\widehat{\mathbf{A}}$ are used in practice as well (such as $\widehat{\mathbf{A}} = \widetilde{\mathbf{D}}^{-1} \widetilde{\mathbf{A}}$).

## 4 METHOD

The proposed *graph convolutional normalizing flow* (GC-Flow) extends a usual normalizing flow acting on data points separately to one that acts on all graph nodes collectively. Following the notations used in Section 3.3, starting with $\mathbf{X}^{(0)} \equiv \mathbf{X}$, where $\mathbf{X}$ is an $n \times D$ input feature matrix for all $n$ nodes in the graph, we define a GC-Flow $\mathbf{F}(\mathbf{X}) : \mathbb{R}^{n \times D} \to \mathbb{R}^{n \times D}$ that is a composition of $T$ constituent flows $\mathbf{F} = \mathbf{F}_T \circ \mathbf{F}_{T-1} \circ \cdots \circ \mathbf{F}_1$, where each constituent flow $\mathbf{F}_i$ computes

$$\mathbf{X}^{(i)} = \mathbf{F}_i(\underbrace{\widehat{\mathbf{A}} \mathbf{X}^{(i-1)}}_{\widetilde{\mathbf{X}}^{(i)}}), \quad i = 1, \ldots, T. \quad (4)$$

The final representation of the nodes is the matrix $\mathbf{Z} \equiv \mathbf{X}^{(T)}$.

*GC-Flow is a normalizing flow.* Similar to other normalizing flows, each constituent flow preserves the feature dimension; that is, each $\mathbf{F}_i$ is an $\mathbb{R}^{n \times D} \to \mathbb{R}^{n \times D}$ function. Furthermore, we let $\mathbf{F}_i$ act on each row of the input argument $\widetilde{\mathbf{X}}^{(i)}$ separately and identically. In other words, from the functionality perspective, $\mathbf{F}_i$ can be equivalently replaced by some function $\mathbf{f}_i : \mathbb{R}^{1 \times D} \to \mathbb{R}^{1 \times D}$ that computes $\mathbf{x}_j^{(i)} = \mathbf{f}_i(\widetilde{\mathbf{x}}_j^{(i)})$ for a node $j$. The main difference between GC-Flow and a usual flow is that the input argument of $\mathbf{f}_i$ contains not only the information of node $j$ but also that of its

neighbors. One may consider a usual flow to be a special case of GC-Flows, when $\widehat{\mathbf{A}} = \mathbf{I}$ (e.g., the graph contains no edges).

*Moreover, GC-FLow is a GNN.* In particular, a constituent flow $\mathbf{F}_i$ of (4) resembles a GCN layer of (3) by making use of graph convolutions—multiplying $\widehat{\mathbf{A}}$ to the flow/layer input $\mathbf{X}^{(i-1)}$. When $\widehat{\mathbf{A}}$ results from the normalization defined by GCN, such a graph convolution approximates a low-pass filter (Kipf & Welling, 2017). In a sense, the GC-Flow architecture is more general than a GCN architecture, because one may interpret the dense layer (represented by the parameter matrix $\mathbf{W}^{(i-1)}$) followed by a nonlinear activation $\sigma_i$ in (3) as an example of the constituent flow $\mathbf{F}_i$ in (4). However, such a conceptual connection needs a few adjustments to make a GC-Flow and a GCN mathematically equivalent, because $\mathbf{W}^{(i-1)}$ in GCN is not required to preserve the feature dimension and $\sigma_i$ of GCN has a zero derivative on the negative axis, compromising invertibility. The nearest adjustment can be made via using the *Sylvester flow* (van den Berg et al., 2018), which adds a residual connection and uses an additional parameter matrix $\mathbf{U}^{(i-1)}$ to preserve the feature dimension:[1] $\mathbf{X}^{(i)} = \mathbf{X}^{(i-1)} + \sigma_i(\widehat{\mathbf{A}}\mathbf{X}^{(i-1)}\mathbf{W}^{(i-1)})\mathbf{U}^{(i-1)}$. However, the Sylvester flow generally has a limited capacity (Kobyzev et al., 2021) and a more sophisticated flow is instead used as $\mathbf{F}_i$, such as the affine coupling layer introduced in Section 3.1.

A major distinction between the GC-Flow and a usual GNN lies in the training objective. To encourage a good clustering structure of the representation $\mathbf{Z}$, we use a maximum-likelihood kind of objective for all graph nodes, because it is equivalent to maximizing the likelihood that $\mathbf{Z}$ forms a Gaussian mixture:

$$\max \mathcal{L} := \frac{1-\lambda}{|\mathcal{D}_l|} \sum_{(\mathbf{x}, y=k) \in \mathcal{D}_l} \log p(\mathbf{x}, y=k) + \frac{\lambda}{|\mathcal{D}_u|} \sum_{\mathbf{x} \in \mathcal{D}_u} \log p(\mathbf{x}), \tag{5}$$

where $\mathcal{D}_l$ denotes the set of labeled nodes, $\mathcal{D}_u$ denotes the set of unlabeled nodes, and $\lambda \in (0, 1)$ is a tunable hyperparameter balancing labeled and unlabeled information. It is useful to compare $\mathcal{L}$ with the usual (negative) cross-entropy loss for training GNNs. First, for training a usual GNN, no loss is incurred on the unlabeled nodes, because their likelihoods are not modeled. Second, for a labeled node $\mathbf{x}$ with true label $k$, the negative cross-entropy is $\log p(y = k|\mathbf{x})$, while the likelihood term over labeled data in (5) is a joint probability of $\mathbf{x}$ and $y$: $\log p(\mathbf{x}, y = k) = \log p(y = k|\mathbf{x}) + \log p(\mathbf{x})$. Fundamentally, GC-Flow belongs to the class of generative classification models, while GNNs belong to the class of discriminative models. Under Bayesian paradigm, the former models the class prior and the class conditional likelihood, while the latter models only the posterior.

In what follows, we will define the proposed probability model $p(\mathbf{x}|y)p(y)$ for a node $\mathbf{x}$, so that the loss $\mathcal{L}$ can be computed and the label $y$ can be predicted via $\arg\max_k p(y = k|\mathbf{x})$. We first need an important lemma on the Jacobian determinant when a graph convolution is involved in the flow.

### 4.1 DETERMINANT LEMMA

The Jacobian determinant of each constituent flow $\mathbf{F}_i$ defined in (4) is needed for training a GC-Flow. The Jacobian is an $nD \times nD$ matrix, but it admits a special block structure that allows the determinant to be computed as a product of determinants on $D$ matrices of size $n \times n$, after rearrangement of the QR factorization factors of the Jacobians of $\mathbf{f}_i$. The following lemma summarizes this finding; the proof is given in the appendix. For notational convenience, we remove the flow index and use $\mathbf{G}$ to denote a generic constituent flow.

**Lemma 1.** *Let $\mathbf{X} \in \mathbb{R}^{n \times D}$ and $\widehat{\mathbf{A}} \in \mathbb{R}^{n \times n}$. Let $\mathbf{Y} = \mathbf{G}(\widetilde{\mathbf{X}})$, where $\widetilde{\mathbf{X}} \equiv \widehat{\mathbf{A}}\mathbf{X}$ and $\mathbf{G} : \mathbb{R}^{n \times D} \to \mathbb{R}^{n \times D}$ acts on each row of the input matrix independently and identically. Let $\mathbf{g} : \mathbb{R}^D \to \mathbb{R}^D$ be functionally equivalent to $\mathbf{G}$; that is, $\mathbf{y}_i = \mathbf{g}(\widetilde{\mathbf{x}}_i)$ where $\mathbf{y}_i$ and $\widetilde{\mathbf{x}}_i$ are the $i$-th row of $\mathbf{Y}$ and $\widetilde{\mathbf{X}}$, respectively. Then, $\left| \det\left( \frac{d\mathbf{Y}}{d\mathbf{X}} \right) \right| = |\det \widehat{\mathbf{A}}|^D \prod_{i=1}^n |\det \nabla \mathbf{g}(\widetilde{\mathbf{x}}_i)|$.*

Putting back the flow index, the above lemma suggests that, by the chain rule, the Jacobian determinant of the entire GC-Flow $\mathbf{F}$ is

$$|\det \nabla \mathbf{F}(\mathbf{X})| = |\det \widehat{\mathbf{A}}|^{TD} \prod_{j=1}^T \prod_{i=1}^n |\det \nabla \mathbf{f}_j(\widetilde{\mathbf{x}}_i^{(j)})|. \tag{6}$$

---

[1]For notational convenience and consistency with the GNN literature, here we omit the often-used bias term.

Note that to maintain invertibility of the flow, the matrix $\widehat{\mathbf{A}}$ must be nonsingular. We will define the probability model for GC-Flow based on equality (6).

## 4.2 PROBABILITY MODEL

Different from a usual normalizing flow, where the representation $\mathbf{z}_i$ for the $i$-th data point depends on its input feature vector $\mathbf{x}_i$, in a GC-Flow, $\mathbf{z}_i$ depends on (a possibly substantial portion of) the entire node set $\mathbf{X}$, because of the $\widehat{\mathbf{A}}$-multiplication. To this end, we use $p(\mathbf{X})$ and $\pi(\mathbf{Z})$ to denote the joint distribution of the node feature vectors and that of the representations, respectively. We still have, by the change-of-variable formula,

$$p(\mathbf{X}) = \pi(\mathbf{Z})|\det \nabla \mathbf{F}(\mathbf{X})|, \tag{7}$$

where the Jacobian determinant has been derived in (6). Under the freedom of modeling and for convenience, we opt to let $\pi(\mathbf{Z})$ be expressed as $\pi(\mathbf{Z}) = \pi(\mathbf{z}_1)\pi(\mathbf{z}_2)\cdots\pi(\mathbf{z}_n)$, where each $\pi(\mathbf{z}_i)$ is an independent and identically distributed Gaussian mixture (2). Similarly, we assume the nodes to be independent to start with; that is, $p(\mathbf{X}) = p(\mathbf{x}_1)p(\mathbf{x}_2)\cdots p(\mathbf{x}_n)$.

For generative modeling, a task is to model the class prior $p(y)$ and the class conditional likelihood $p(\mathbf{x}|y)$, such that the posterior prediction model $p(y|\mathbf{x})$ can be easily obtained as proportional to $p(\mathbf{x}|y)p(y)$, by Bayes' Theorem. To this end, we define

$$p(\mathbf{x}_i|y_i = k) := \mathcal{N}(\mathbf{z}_i; \boldsymbol{\mu}_k, \boldsymbol{\Sigma}_k)|\det \widehat{\mathbf{A}}|^{TD/n} \prod_{j=1}^{T} |\det \nabla \mathbf{f}_j(\widetilde{\mathbf{x}}_i^{(j)})| \quad \text{and} \quad p(y_i = k) = \phi_k. \tag{8}$$

Such a definition is self-consistent. First, marginalizing over the label $y_i$ and using the Gaussian mixture definition (2) for $\pi(\mathbf{z}_i)$, we obtain the marginal likelihood

$$p(\mathbf{x}_i) = \pi(\mathbf{z}_i)|\det \widehat{\mathbf{A}}|^{TD/n} \prod_{j=1}^{T} |\det \nabla \mathbf{f}_j(\widetilde{\mathbf{x}}_i^{(j)})|. \tag{9}$$

Then, by the modeling of $\pi(\mathbf{Z})$ and $p(\mathbf{X})$, taking the product for all nodes and using the Jacobian determinant formula derived in (6), we exactly recover the density formula (7). We will use (8) and (9) to compute the labeled part and the unlabeled part of the loss (5), respectively.

The modeling of $\pi(\mathbf{Z})$ as a product of $\pi(\mathbf{z}_i)$'s reflects independence, which may seem conceptually at odds with graph convolutions, where a node's representation depends on the information of nodes in it's $T$-hop neighborhood. However, nothing prevents the convolution results to be independent, just like the case that a usual normalizing flow can decorrelate the input features and make each transformed feature independent, when postulating a standard normal distribution output. It is the aim of the independence of the $\mathbf{z}_i$'s that enables finding the most probable GC-Flow.

## 4.3 TRAINING AND COSTS

Despite inheriting the generative characteristics of FlowGMMs (including the training loss), GC-Flows are by nature a GNN, because the graph convolution operation ($\widehat{\mathbf{A}}$-multiplication) involves a node's neighbor set when computing the output of a constituent flow for this node. Due to space limitation, we discuss the complication of training and inference owing to neighborhood explosion in Appendix C; these discussions share great similarities with the GNN case. Additionally, we compare the full-batch training costs of GC-Flow and GCN in Appendix D, which suggests that they are comparable and admit the same scaling behavior.

## 4.4 IMPROVING PERFORMANCE THROUGH PARAMETERIZING $\widehat{\mathbf{A}}$

So far, we have treated $\widehat{\mathbf{A}}$ as the normalization of the graph adjacency matrix $\mathbf{A}$ defined by GCN (see Section 3.3). One convenience of doing so is that $\det \widehat{\mathbf{A}}$ is a constant and can be safely omitted in the loss calculation. One may improve the quality of GC-Flow through introducing parameterizations to $\widehat{\mathbf{A}}$. One approach, which we call GC-Flow-p, is to parameterize the edge weights. This approach is similar to GAT (Veličković et al., 2018) that uses attention weights to redefine $\widehat{\mathbf{A}}$. Another approach, which we call GC-Flow-l, is to learn $\widehat{\mathbf{A}}$ in its entirety without resorting to the (possibly unknown) graph structure. For this purpose, several approaches have been developed; see, e.g., Franceschi et al. (2019); Wu et al. (2020); Shang et al. (2021); Fatemi et al. (2021); Dai & Chen (2022).

Table 1: Comparison of GMM-based generative models, GNN-based discriminative models, and GC-Flow for semi-supervised classification and clustering. Standard deviations are obtained by repeating model training ten times. For each data set and metric, the two best cases are boldfaced.

| | Cora | | Citeseer | | Pubmed | |
|---|---|---|---|---|---|---|
| | Silhouette | Micro-F1 | Silhouette | Micro-F1 | Silhouette | Micro-F1 |
| FlowGMM | $\mathbf{0.739 \pm 0.015}$ | $0.504 \pm 0.021$ | $\mathbf{0.609 \pm 0.034}$ | $0.512 \pm 0.044$ | $\mathbf{0.653 \pm 0.031}$ | $0.734 \pm 0.014$ |
| GMM on $\mathbf{X}$ | $0.162 \pm 0.000$ | $0.163 \pm 0.000$ | $0.071 \pm 0.000$ | $0.085 \pm 0.000$ | $0.062 \pm 0.000$ | $0.581 \pm 0.000$ |
| GMM on $\widehat{\mathbf{A}}\mathbf{X}$ | $0.144 \pm 0.000$ | $0.173 \pm 0.000$ | $0.089 \pm 0.000$ | $0.182 \pm 0.000$ | $0.183 \pm 0.000$ | $0.411 \pm 0.000$ |
| GCN | $0.340 \pm 0.003$ | $0.813 \pm 0.007$ | $0.314 \pm 0.016$ | $0.700 \pm 0.025$ | $0.453 \pm 0.006$ | $\mathbf{0.791 \pm 0.004}$ |
| GraphSAGE | $0.346 \pm 0.004$ | $0.801 \pm 0.005$ | $0.278 \pm 0.007$ | $0.697 \pm 0.007$ | $0.440 \pm 0.018$ | $0.769 \pm 0.011$ |
| GAT | $0.383 \pm 0.003$ | $\mathbf{0.825 \pm 0.005}$ | $0.304 \pm 0.003$ | $\mathbf{0.702 \pm 0.007}$ | $0.435 \pm 0.010$ | $0.774 \pm 0.005$ |
| GC-Flow | $\mathbf{0.734 \pm 0.006}$ | $\mathbf{0.815 \pm 0.011}$ | $\mathbf{0.538 \pm 0.022}$ | $\mathbf{0.714 \pm 0.011}$ | $\mathbf{0.669 \pm 0.021}$ | $\mathbf{0.791 \pm 0.009}$ |

| | Computers | | Photo | | Wiki-CS | |
|---|---|---|---|---|---|---|
| | Silhouette | Micro-F1 | Silhouette | Micro-F1 | Silhouette | Micro-F1 |
| FlowGMM | $\mathbf{0.540 \pm 0.024}$ | $0.614 \pm 0.026$ | $\mathbf{0.704 \pm 0.027}$ | $0.599 \pm 0.089$ | $\mathbf{0.677 \pm 0.011}$ | $0.671 \pm 0.011$ |
| GMM on $\mathbf{X}$ | $-0.018 \pm 0.00$ | $0.102 \pm 0.000$ | $-0.024 \pm 0.00$ | $0.120 \pm 0.000$ | $0.088 \pm 0.000$ | $0.124 \pm 0.000$ |
| GMM on $\widehat{\mathbf{A}}\mathbf{X}$ | $-0.021 \pm 0.00$ | $0.062 \pm 0.000$ | $-0.041 \pm 0.00$ | $0.098 \pm 0.000$ | $0.026 \pm 0.000$ | $0.188 \pm 0.000$ |
| GCN | $0.357 \pm 0.026$ | $0.812 \pm 0.016$ | $0.388 \pm 0.003$ | $0.891 \pm 0.012$ | $0.264 \pm 0.005$ | $\mathbf{0.775 \pm 0.005}$ |
| GraphSAGE | $0.434 \pm 0.030$ | $0.761 \pm 0.024$ | $0.386 \pm 0.007$ | $0.839 \pm 0.020$ | $0.233 \pm 0.009$ | $0.771 \pm 0.003$ |
| GAT | $0.431 \pm 0.015$ | $\mathbf{0.814 \pm 0.023}$ | $0.425 \pm 0.020$ | $\mathbf{0.900 \pm 0.009}$ | $0.278 \pm 0.008$ | $0.773 \pm 0.003$ |
| GC-Flow | $\mathbf{0.487 \pm 0.012}$ | $\mathbf{0.847 \pm 0.007}$ | $\mathbf{0.655 \pm 0.013}$ | $\mathbf{0.917 \pm 0.004}$ | $\mathbf{0.717 \pm 0.010}$ | $\mathbf{0.775 \pm 0.002}$ |

In a later experiment, we will give examples for GC-Flow-p and GC-Flow-l (see Appendix B for details) and investigate the performance improvement over GC-Flow. Note that the parameterization may lead to a different $\widehat{\mathbf{A}}$ for each constituent flow. Hence, we add the flow index $^{(j)}$ to $\widehat{\mathbf{A}}$ and rewrite the Jacobian determinant (6) as $|\det \nabla \mathbf{F}(\mathbf{X})| = \prod_{j=1}^{T} \left( |\det \widehat{\mathbf{A}}^{(j)}|^{D} \prod_{i=1}^{n} |\det \nabla \mathbf{f}_j(\widetilde{\mathbf{x}}_i^{(j)})| \right)$. The probability models (8) and (9) are correspondingly rewritten, respectively, as

$$p(\mathbf{x}_i|y_i = k) := \mathcal{N}(\mathbf{z}_i; \boldsymbol{\mu}_k, \boldsymbol{\Sigma}_k) \prod_{j=1}^{T} |\det \widehat{\mathbf{A}}^{(j)}|^{D/n} |\det \nabla \mathbf{f}_j(\widetilde{\mathbf{x}}_i^{(j)})| \quad \text{and} \quad p(y_i = k) = \phi_k,$$

$$p(\mathbf{x}_i) = \pi(\mathbf{z}_i) \prod_{j=1}^{T} |\det \widehat{\mathbf{A}}^{(j)}|^{D/n} |\det \nabla \mathbf{f}_j(\widetilde{\mathbf{x}}_i^{(j)})|.$$

These two formulas are used to substitute the labeled and unlabeled parts of the loss (5), respectively.

## 5 EXPERIMENTS

In this section, we conduct a comprehensive set of experiments to evaluate the performance of GC-Flow on graph data and demonstrate that it is competitive with GNNs for classification, while being advantageous in learning representations that extract the clustering structure of the data.

**Data sets.** We use six benchmark node-classification data sets. Data sets **Cora**, **Citeseer**, and **Pubmed** are citation graphs, where each node is a document and each edge represents the citation relation between two documents. We follow the predefined splits in Kipf & Welling (2017). Data sets **Computers** and **Photo** are subgraphs of the Amazon co-purchase graph (McAuley et al., 2015). They do not have a predefined split. We randomly sample 200/1300/1000 nodes for training/validation/testing for Computers and 80/620/1000 for Photo. The data set **Wiki-CS** is a web graph where nodes are Wikipedia articles and edges are hyperlinks (Mernyei & Cangea, 2020). We use one of the predefined splits. For statistics of the data sets, see Table 4 in Appendix E.

**Baselines.** We compare GC-Flow with both discriminative and generative models. For discriminative models, we use three widely used GNNs: GCN, GraphSAGE, and GAT. For generative models, besides FlowGMM, we use the basic Gaussian mixture model (GMM). GMM is not parameterized; it takes either the node features $\mathbf{X}$ or the graph-transformed features $\widetilde{\mathbf{X}} = \widehat{\mathbf{A}}\mathbf{X}$ as input.

**Metrics.** For measuring classification quality, we use the standard micro-averaged F1 score. For evaluating clustering, we mainly use the silhouette coefficient. This metric does not require ground-truth cluster labels and it measures the separation of clusters. Better separation indicates a more interpretable structure. We additionally use NMI (normalized mutual information) and ARI (adjusted rand index) to measure clustering quality with known ground truths.

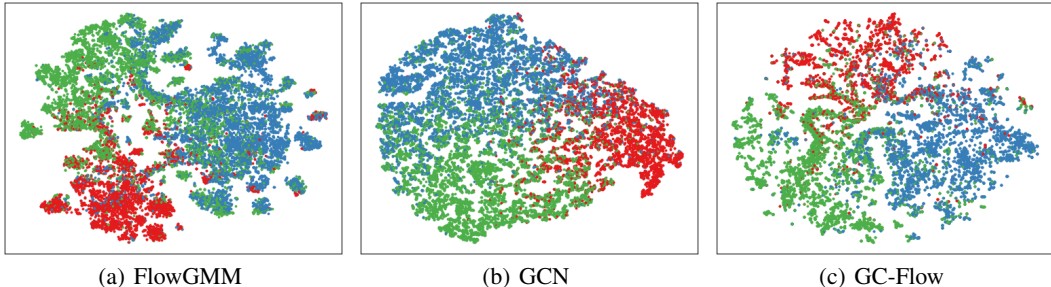

| (a) FlowGMM | (b) GCN | (c) GC-Flow |

Figure 2: Representation space of the data set Pubmed under different models.

**Implementation details and hyperparameter information** may be found in Appendix E.

**Classification and clustering performance.** Table 1 lists the F1 scores and the silhouette coefficients for all data sets and all compared models. GNNs are always better than GMMs for classification; while the flow version of GMM, FlowGMM, beats all GNNs on cluster separation. Our model, GC-Flow, is competitive with the better of the two and is always the best or the second best. When being the best, some of the improvements are rather substantial, such as the F1 score for Computers and the silhouette coefficient for Wiki-CS. It is interesting to note that the basic GMMs perform rather poorly. This phenomenon is not surprising, because without any neural network parameterization, they cannot compete with other models that allow feature transformations to encourage class separation or cluster separation.

To further illustrate the clustering quality of GC-Flow, we compare it with several contrastive-based methods that produce competitive clusterings: DGI (Veličković et al., 2019), GRACE (Zhu et al., 2020), GCA (Zhu et al., 2021), GraphCL (You et al., 2020), and MVGRL (Hassani & Khasahmadi, 2020). Table 2 lists the results for

Table 2: Clustering performance of various GNN methods. The two best cases are boldfaced. Data set: Cora.

|  | NMI | ARI | Silhouette |
|---|---|---|---|
| DGI | $0.592 \pm 0.001$ | $0.570 \pm 0.002$ | $0.330 \pm 0.000$ |
| GRACE | $0.475 \pm 0.028$ | $0.394 \pm 0.047$ | $0.153 \pm 0.011$ |
| GCA | $0.418 \pm 0.053$ | $0.259 \pm 0.058$ | $0.301 \pm 0.005$ |
| GraphCL | $0.577 \pm 0.002$ | $0.482 \pm 0.003$ | $0.297 \pm 0.002$ |
| MVGRL | $\mathbf{0.612 \pm 0.026}$ | $\mathbf{0.576 \pm 0.062}$ | $\mathbf{0.369 \pm 0.013}$ |
| GC-Flow | $\mathbf{0.621 \pm 0.013}$ | $\mathbf{0.631 \pm 0.008}$ | $\mathbf{0.734 \pm 0.006}$ |

Cora and Table 5 in Appendix F includes more data sets. For Cora, GC-Flow delivers the best performance on all metrics, with a silhouette score more than double of the second best. Compared with NMI and ARI, silhouette is a metric that takes no knowledge of the ground truth but measures solely the cluster separation in space. This result suggests that the clusters obtained from GC-Flow are more structurally separated, albeit improving less the cluster agreement.

**Training behavior.** Figure 5 (see Appendix F) plots the convergence behavior of the training loss for FlowGMM, GCN, and GC-Flow. All methods converge favorably, with GCN reaching the plateau earlier, while FlowGMM and GC-Flow converge at a rather similar speed.

**Visualization of the representation space.** To complement the numerical metrics, we visualize the representation space of FlowGMM, GCN, and GC-Flow by using a t-SNE plot (Van der Maaten & Hinton, 2008), for qualitative evaluation. The representations for FlowGMM and GC-Flow are the $z_i$'s, while those for GCN are extracted from the penultimate activations. The results for Cora are given earlier in Figure 1; we additionally give the results for Pubmed in Figure 2. From both figures, one sees that similar to FlowGMM, GC-Flow exhibits a better clustering structure than does GCN, which produces little separation for the data. More visualizations are provided in Appendix F.

**Analysis on depth.** Figure 3 plots the performance of FlowGMM, GCN, and GC-Flow as the number of layers/flows increases. One sees that the classification performance of GCN deteriorates with more layers, in agreement with the well-known oversmoothing phenomenon (Li et al., 2018), while the clustering performance is generally stable. On the other hand, the classification performance of FlowGMM and GC-Flow does not show a unique pattern: for Cora, it degrades, while for Pubmed, it stabilizes. The clustering performance of FlowGMM and GC-Flow generally degrades, except for the curious case of GC-Flow on Cora, where the silhouette coefficient shows a V-shape. Nevertheless, generally a smaller depth is preferred for all models.

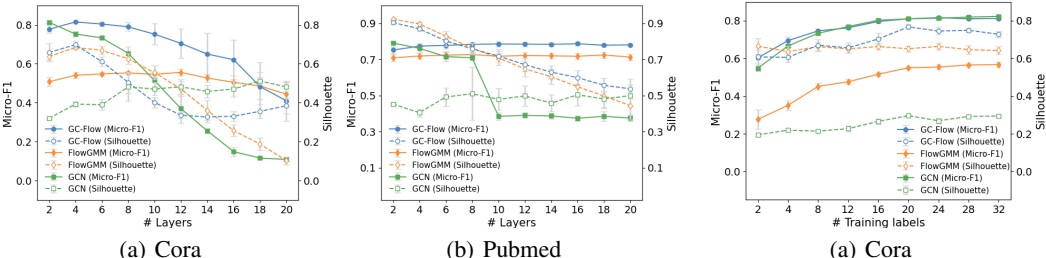

(a) Cora          (b) Pubmed          (a) Cora

Figure 3: Performance variation with respect to the network depth/number of flows.

Figure 4: Performance variation with respect to the labeling rate.

Table 3: Effect of modeling $\widehat{\mathbf{A}}$. Boldfaced numbers indicate improvement over GC-Flow.

|  | Pubmed | | Computers | | Photo | |
|---|---|---|---|---|---|---|
|  | Silhouette | Micro-F1 | Silhouette | Micro-F1 | Silhouette | Micro-F1 |
| GC-Flow | $0.669 \pm 0.021$ | $0.791 \pm 0.009$ | $0.487 \pm 0.012$ | $0.847 \pm 0.007$ | $0.655 \pm 0.013$ | $0.917 \pm 0.004$ |
| GC-Flow-p | $\mathbf{0.804 \pm 0.010}$ | $0.790 \pm 0.007$ | $\mathbf{0.706 \pm 0.019}$ | $0.841 \pm 0.006$ | $\mathbf{0.874 \pm 0.011}$ | $0.914 \pm 0.008$ |
| GC-Flow-l | $\mathbf{0.856 \pm 0.029}$ | $0.783 \pm 0.009$ | $\mathbf{0.582 \pm 0.020}$ | $\mathbf{0.851 \pm 0.009}$ | $\mathbf{0.842 \pm 0.006}$ | $0.911 \pm 0.005$ |

**Analysis on labeling rate.** Figure 4 plots the performance of FlowGMM, GCN, and GC-Flow as the number of training labels per class increases. One sees that for all models, the performance generally improves with more labeled data. The improvement is more steady and noticeable for classification, while being less significant for clustering. Additionally, GC-Flow classifies significantly better than does GCN at the low-labeling rate regime, achieving a 10.04% relative improvement in the F1 score when there are only two labeled nodes per class.

**Improving performance with additional parameterization.** We experiment with two variants of GC-Flow by introducing parameterizations to $\widehat{\mathbf{A}}$. The variant GC-Flow-p uses an idea similar to GAT, through embedding flow inputs and computing an additive attention on the graph edges to redefine their weights. Another variant GC-Flow-l also computes weights, but rather than using them to define $\widehat{\mathbf{A}}$, it treats each weight as a probability of edge presence and samples the corresponding Bernoulli distribution to obtain a binary sample $\widehat{\mathbf{A}}$. The details are given in Appendix B.

Table 3 lists the performance of GC-Flow-p and GC-Flow-l on three selected data sets, where the improvement over GC-Flow is notable. The improvement predominantly appears for clustering, with the most striking increase from 0.487 to 0.706. The increase of silhouette coefficients generally come with a marginal decrease in the F1 score, but the decrement amount is below the standard deviation. In one occasion (Computers), the F1 score even increases, despite also being marginal.

## 6 CONCLUSIONS

We have developed a generative GNN model which, rather than directly computing the class posterior $p(y|\mathbf{x})$, computes the class conditional likelihood $p(\mathbf{x}|y)$ and applies the Bayes rule together with the class prior $p(y)$ for prediction. A benefit of such a model is that one may control the representation of the data (e.g., a clustering structure) through modeling the representation distribution (e.g., optimizing it toward a mixture of well-separated unimodal distributions). We achieve so by designing the GNN as a normalizing flow that additionally incorporates graph convolutions. Interestingly, the graph adjacency matrix appears in the density computation of the normalizing flow as a stand-alone term, which could be ignored if it is a constant, or easily optimized if it is parameterized. We demonstrate that the proposed model not only maintains the predictive power of the past GNNs, but also produces high-quality clusters in the representation space.

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

## A PROOF OF LEMMA 1

The Jacobian $\frac{d\mathbf{Y}}{d\mathbf{X}}$ is an $nD \times nD$ matrix. By the chain rule, an entry of it is $\frac{d\mathbf{Y}_{ij}}{d\mathbf{X}_{pq}} = \widehat{\mathbf{A}}_{ip}\mathbf{J}^i_{jq}$, where $\mathbf{J}^i := \nabla\mathbf{g}(\widetilde{\mathbf{x}}_i) \in \mathbb{R}^{D \times D}$. Hence, $\frac{d\mathbf{Y}}{d\mathbf{X}}$ can be expressed as the following block matrix (up to permutation and transpose that do not affect the absolute value of the determinant):

$$\begin{bmatrix} \widehat{\mathbf{A}}_{11}\mathbf{J}^1 & \widehat{\mathbf{A}}_{12}\mathbf{J}^1 & \cdots & \widehat{\mathbf{A}}_{1n}\mathbf{J}^1 \\ \widehat{\mathbf{A}}_{21}\mathbf{J}^2 & \widehat{\mathbf{A}}_{22}\mathbf{J}^2 & \cdots & \widehat{\mathbf{A}}_{2n}\mathbf{J}^2 \\ \vdots & \vdots & \ddots & \vdots \\ \widehat{\mathbf{A}}_{n1}\mathbf{J}^n & \widehat{\mathbf{A}}_{n2}\mathbf{J}^n & \cdots & \widehat{\mathbf{A}}_{nn}\mathbf{J}^n \end{bmatrix}.$$

Since any matrix admits a QR factorization, let $\mathbf{J}^i = \mathbf{Q}^i\mathbf{R}^i$ for all $i$, where $\mathbf{Q}^i$ is unitary and $\mathbf{R}^i$ is upper-triangular. Then, the above block matrix is equal to

$$\begin{bmatrix} \mathbf{Q}^1 & & & \\ & \mathbf{Q}^2 & & \\ & & \ddots & \\ & & & \mathbf{Q}^n \end{bmatrix} \begin{bmatrix} \widehat{\mathbf{A}}_{11}\mathbf{R}^1 & \widehat{\mathbf{A}}_{12}\mathbf{R}^1 & \cdots & \widehat{\mathbf{A}}_{1n}\mathbf{R}^1 \\ \widehat{\mathbf{A}}_{21}\mathbf{R}^2 & \widehat{\mathbf{A}}_{22}\mathbf{R}^2 & \cdots & \widehat{\mathbf{A}}_{2n}\mathbf{R}^2 \\ \vdots & \vdots & \ddots & \vdots \\ \widehat{\mathbf{A}}_{n1}\mathbf{R}^n & \widehat{\mathbf{A}}_{n2}\mathbf{R}^n & \cdots & \widehat{\mathbf{A}}_{nn}\mathbf{R}^n \end{bmatrix}.$$

Because left block matrix is unitary, it does not change the determinant. Hence, we only need to compute the determinant of the right block matrix. We may rearrange this matrix into the following form, while maintaining the absolute value of the determinant:

$$\begin{bmatrix} \widehat{\mathbf{A}} \odot \mathbf{S}^{11} & \widehat{\mathbf{A}} \odot \mathbf{S}^{12} & \cdots & \widehat{\mathbf{A}} \odot \mathbf{S}^{1n} \\ \widehat{\mathbf{A}} \odot \mathbf{S}^{21} & \widehat{\mathbf{A}} \odot \mathbf{S}^{22} & \cdots & \widehat{\mathbf{A}} \odot \mathbf{S}^{2n} \\ \vdots & \vdots & \ddots & \vdots \\ \widehat{\mathbf{A}} \odot \mathbf{S}^{n1} & \widehat{\mathbf{A}} \odot \mathbf{S}^{n2} & \cdots & \widehat{\mathbf{A}} \odot \mathbf{S}^{nn} \end{bmatrix} \quad \text{where} \quad \mathbf{S}^{ij} = \begin{bmatrix} \mathbf{R}^1_{ij} & \mathbf{R}^1_{ij} & \cdots & \mathbf{R}^1_{ij} \\ \mathbf{R}^2_{ij} & \mathbf{R}^2_{ij} & \cdots & \mathbf{R}^2_{ij} \\ \vdots & \vdots & \ddots & \vdots \\ \mathbf{R}^n_{ij} & \mathbf{R}^n_{ij} & \cdots & \mathbf{R}^n_{ij} \end{bmatrix} \text{ for all } i, j \text{ pairs.}$$

Because each $\mathbf{R}^k$ is upper-triangular, the matrix $\mathbf{S}^{ij}$ is zero whenever $i > j$. Therefore, the block matrix above is block upper-triangular and its absolute determinant is equal to

$$\prod_{i=1}^{D} |\det(\widehat{\mathbf{A}} \odot \mathbf{S}^{ii})|.$$

Note that $\mathbf{S}^{ii}$ is a matrix with identical rows, for each $i$. Then, by the definition of determinant (see the Leibniz formula), $\det(\widehat{\mathbf{A}} \odot \mathbf{S}^{ii}) = \det(\widehat{\mathbf{A}}) \cdot \mathbf{R}^1_{ii}\mathbf{R}^2_{ii}\cdots\mathbf{R}^n_{ii}$. Therefore,

$$\prod_{i=1}^{D} |\det(\widehat{\mathbf{A}} \odot \mathbf{S}^{ii})| = |\det\widehat{\mathbf{A}}|^D \cdot |\mathbf{R}^1_{11}\mathbf{R}^2_{11}\cdots\mathbf{R}^n_{11}\mathbf{R}^1_{22}\mathbf{R}^2_{22}\cdots\mathbf{R}^n_{22}\cdots\mathbf{R}^1_{DD}\mathbf{R}^2_{DD}\cdots\mathbf{R}^n_{DD}|$$

$$= |\det\widehat{\mathbf{A}}|^D \cdot |\det\mathbf{J}^1| \cdot |\det\mathbf{J}^2|\cdots|\det\mathbf{J}^n|,$$

which concludes the proof.

## B PARAMETERIZATIONS OF $\widehat{\mathbf{A}}$

With parameterization, $\widehat{\mathbf{A}}$ may differ in the constituent flows. Hence, we use the flow index $j$ to distinguish them; i.e., $\widehat{\mathbf{A}}^{(j)}$. Let a graph be denoted by $\mathcal{G} = (\mathcal{V}, \mathcal{E})$, where $\mathcal{V}$ is the node set and $\mathcal{E}$ is the edge set.

### B.1 GC-FLOW-P VARIANT

The GC-Flow-p variant parameterizes $\widehat{\mathbf{A}}^{(j)}$ by using an idea similar to GAT (Veličković et al., 2018), where an existing edge $(i, k) \in \mathcal{E}$ is reweighted by using attention scores. Let $\mathbf{E}_1, \mathbf{E}_2 : \mathbb{R}^D \to \mathbb{R}^d$ be two embedding networks that map a $D$-dimensional flow input to a $d$-dimensional vector, and let $\mathbf{M} : \mathbb{R}^{2d} \to \mathbb{R}^1$ be a feed-forward network. We compute two sets of vectors

$$\mathbf{h}_i^{(j-1)} = \text{ReLU}\left(\mathbf{E}_1(\mathbf{x}_i^{(j-1)})\right), \quad \mathbf{g}_k^{(j-1)} = \text{ReLU}\left(\mathbf{E}_2(\mathbf{x}_k^{(j-1)})\right)$$

and concatenate them to compute a pre-attention coefficient $\alpha_{ik}^{(j)}$ for all $(i, k) \in \mathcal{E}$:

$$\alpha_{ik}^{(j)} = \mathbf{M}\Big( [\mathbf{h}_i^{(j-1)} \,\|\, \mathbf{g}_k^{(j-1)}] \Big).$$

Then, constructing a matrix $\mathbf{S}^{(j)}$ where

$$\mathbf{S}_{ik}^{(j)} = \begin{cases} \text{LeakyReLU}(\alpha_{ik}^{(j)}) & \text{if} (i, k) \in \mathcal{E}, \\ -\infty & \text{otherwise}, \end{cases}$$

we define $\widehat{\mathbf{A}}^{(j)}$ through a row-wise softmax:

$$\widehat{\mathbf{A}}^{(j)} = \text{softmax}(\mathbf{S}^{(j)}).$$

This parameterization differs from GAT mainly in using more complex embedding networks $\mathbf{E}_1$ and $\mathbf{E}_2$ than a single feed-forward layer to compute the vectors $\mathbf{h}_i^{(j-1)}$ and $\mathbf{g}_k^{(j-1)}$. Moreover, we do not use multiple heads.

## B.2  GC-FLOW-L VARIANT

The GC-Flow-l variant learns a new graph structure. For computational efficiency, the learning of the structure is based on the given edge set $\mathcal{E}$; that is, only edges are removed from $\mathcal{E}$ but no edges are inserted outside $\mathcal{E}$. The method follows Luo et al. (2021), which hypothesizes that the existing edge set is noisy and aims at removing the noisy edges.

The basic idea uses a prior work on differentiable sampling (Maddison et al., 2016; Jang et al., 2016), which states that the random variable

$$e = \sigma\Big( \Big(\log \epsilon - \log(1 - \epsilon) + \omega\Big)/\tau \Big) \quad \text{where} \quad \epsilon \sim \text{Uniform}(0, 1) \tag{10}$$

follows a distribution that converges to a Bernoulli distribution with success probability $p = (1 + e^{-\omega})^{-1}$ as $\tau > 0$ tends to zero. Hence, we if parameterize $\omega$ and specify that the presence of an edge between a pair of nodes has probability $p$, then using $e$ computed from (10) to fill the corresponding entry of $\widehat{\mathbf{A}}$ will produce a matrix $\widehat{\mathbf{A}}$ that is close to binary. We can use this matrix in GC-Flow, with the hope of improving classification/clustering performance due to the ability of denoising edges. Moreover, because (10) is differentiable with respect to $\omega$, we can train the parameters of $\omega$ like in a usual gradient-based training.

To this end, we let $\mathbf{E}_1, \mathbf{E}_2 : \mathbb{R}^D \to \mathbb{R}^d$ be two embedding networks that embed the pairwise flow inputs as

$$\mathbf{a}_{ik}^{(j)} = \tanh\Big( \mathbf{E}_1\big(\mathbf{x}_i^{(j)}\big) \Big) \odot \tanh\Big( \mathbf{E}_2\big(\mathbf{x}_k^{(j)}\big) \Big), \quad \forall (i, k) \in \mathcal{E}$$

$$\mathbf{b}_{ik}^{(j)} = \tanh\Big( \mathbf{E}_2\big(\mathbf{x}_i^{(j)}\big) \Big) \odot \tanh\Big( \mathbf{E}_1\big(\mathbf{x}_k^{(j)}\big) \Big), \quad \forall (i, k) \in \mathcal{E}$$

where $\mathbf{a}_{ik}^{(j)}, \mathbf{b}_{ik}^{(j)} \in \mathbb{R}^d$ and $\odot$ is the Hadamard product. Then, we take their difference and compute

$$\omega_{ik}^{(j)} = \tanh\Big( \mathbf{1}^T(\mathbf{a}_{ik}^{(j)} - \mathbf{b}_{ik}^{(j)}) \Big),$$

followed by

$$\widehat{e}_{ik}^{(j)} = \sigma\Big( \Big(\log \epsilon - \log(1 - \epsilon) + \omega_{ik}^{(j)}\Big)/\tau \Big) \quad \text{where} \quad \epsilon \sim \text{Uniform}(0, 1),$$

which returns an approximate Bernoulli sample for the edge $(i, k)$. When $\tau$ is not sufficiently close to zero, this sample may not be close enough to binary, and in particular, it is strictly nonzero. To explicitly zero out an edge, we follow Louizos et al. (2017) and introduce two parameters, $\gamma < 0$ and $\xi > 1$, to remove small values of $\widehat{e}_{ik}^{(j)}$:

$$\widehat{\mathbf{A}}_{ik}^{(j)} = \min\Big( 1, \max\big(e_{ik}^{(j)}, 0\big) \Big) \quad \text{where} \quad e_{ik}^{(j)} = \widehat{e}_{ik}^{(j)}(\xi - \gamma) + \gamma.$$

This definition of $\widehat{\mathbf{A}}^{(j)}$ does not insert new edges to the graph (i.e., when $(i, k) \notin \mathcal{E}$, $\widehat{\mathbf{A}}_{ik}^{(j)} = 0$), but only removes (denoises) some edges $(i, k)$ originally in $\mathcal{E}$.

## C  TRAINING AND INFERENCE

Despite inheriting the generative characteristics of FlowGMMs (including the training loss), GC-Flows are by nature a GNN, because the graph convolution operation ($\widehat{\mathbf{A}}$-multiplication) involves a node's neighbor set when computing the output of a constituent flow for this node. Across all constituent flows, the evaluation of the loss on a single node will require the information of the entire $T$-hop neighborhood, causing scalability challenges for large graphs. One may perform full-batch training (the deterministic gradient decent method), which minimizes the multiple evaluations on a node in any constituent flow. Such an approach is the most convenient to implement in the current deep learning frameworks; typical GPU memory can afford handling a medium-scale graph and CPU memory can afford even larger graphs. If one opts to perform mini-batch training (the stochastic gradient decent method), neighborhood sampling for GNNs (e.g., node-wise (Hamilton et al., 2017; Ying et al., 2018), layer-wise (Chen et al., 2018a; Zou et al., 2019), or subgraph-sampling (Chiang et al., 2019; Zeng et al., 2020)) is a popular approach to reducing the computation within the $T$-hop neighborhood.

Inference is faced with the same challenge as training, but since it requires only a single pass on the test set, doing so in full batch or by using large mini-batches may suffice. If one were to use normal mini-batches with neighborhood sampling (for reasons such as being consistent with training), empirical evidence of success has been demonstrated in the GNN literature (Kaler et al., 2022).

## D  COMPLEXITY ANALYSIS

Let us analyze the cost of computing the likelihood loss (5) for GC-Flows. Based on (8) and (9), the cost consists of three parts: that to compute $\widetilde{\mathbf{X}}^{(j)} = \widehat{\mathbf{A}}\mathbf{X}^{(j-1)}$ for each constituent flow indexed by $j$, that to compute the Jacobian determinant $\det \nabla \mathbf{f}_j(\widetilde{\mathbf{x}}_i^{(j)})$ for each node $i$, and that to compute the graph-related determinant $\det \widehat{\mathbf{A}}$. For a fixed graph, the third part is a constant and is omitted in training. The cost of the second part varies according to the type of the flow. If we use an affine-coupling flow as exemplified in Section 3.1, let the cost of the $\mathbf{s}$ and $\mathbf{t}$ networks be $C_{\mathbf{st}}$. Then, the cost of computing the overall loss can be summarized as

$$O\big(\text{nz}(\widehat{\mathbf{A}})DT + nTC_{\mathbf{st}}\big), \tag{11}$$

where $\text{nz}(\widehat{\mathbf{A}})$ denotes the number of nonzeros of $\widehat{\mathbf{A}}$ and recall that $D$ and $T$ are the feature dimension and the number of flows, respectively. An affine-coupling flow can be implemented with varying architectures. For example, for a usual MLP, $C_{\mathbf{st}} = O\big(\sum_{i=0}^{L-1} h_i h_{i+1}\big)$, where $h_0 = \lfloor D/2 \rfloor$; $h_1 \ldots h_{L-1}$ are hidden dimensions; and $h_L = 2\lceil D/2 \rceil$. Note that $O(C_{\mathbf{st}})$ dominates the cost of computing a matrix determinant, which is only $O(D)$, because the Jacobian matrix is triangular in an affine-coupling flow.

It would be useful to compare the above cost with that of the cross-entropy loss for a usual GCN:

$$O\big(\text{nz}(\widehat{\mathbf{A}}) \sum_{j=0}^{T-1} d_j + n \sum_{j=0}^{T-1} d_j d_{j+1}\big), \tag{12}$$

where $d_0 = D$; $d_1 \ldots d_{T-1}$ are hidden dimensions; and $d_T = K$, the number of classes. Here, we assume that the GCN has $T$ layers, comparable to GC-Flow. The two costs (11) and (12) are comparable, part by part. For the first part, $DT$ is comparable to $\sum_{j=0}^{T-1} d_j$, if all the $d_j$'s are similar. In some data sets, the input dimension of GCN is much higher than the hidden and output dimensions, but we correspondingly perform a dimension reduction on the input features when running GC-Flows (as is the practice in the experiments), reducing $DT$ to $D'T$ for some $D' \ll D$. For the second part, the number $L$ of hidden layers in each flow is typically a small number (say, 5), and the hidden dimensions $h_j$'s are comparable to the input dimension $h_0$, rendering comparable terms $TC_{\mathbf{st}}$ versus $\sum_{j=0}^{T-1} d_j d_{j+1}$. Overall, the computational costs of GC-Flow and GCN are similar and their scaling behaviors are the same.

It is worth noting that when one parameterizes $\widehat{\mathbf{A}}$, the cost of computing $\widehat{\mathbf{A}}^{(j)}$ for each flow/layer $j$ will need to be added to the loss computation, for both GC-Flows and GCNs. The cost depends on the specific parameterization and it can be either cheap or expensive. Additionally, GC-Flows require the computation of $\det \widehat{\mathbf{A}}^{(j)}$, whose cost depends on the structure of the matrix, which in turn is determined by the parameterization.

# E  EXPERIMENT DETAILS

**Data sets.** Table 4 summarizes the statistics of the benchmark data sets used in this paper.

Table 4: Data set statistics.

| Data set | # Nodes | # Edges | # Features | # Classes | # Train/val/test |
|---|---|---|---|---|---|
| Cora | 2,708 | 5,429 | 1,433 | 7 | 140 / 500 / 1,000 |
| Citeseer | 3,327 | 4,732 | 3,703 | 6 | 120 / 500 / 1,000 |
| Pubmed | 19,717 | 44,338 | 500 | 3 | 60 / 500 / 1,000 |
| Computers | 13,381 | 245,778 | 767 | 10 | 200 / 1,300 / 1,000 |
| Photo | 7,487 | 119,043 | 745 | 8 | 80 / 620 / 1,000 |
| Wiki-CS | 11,701 | 216,123 | 300 | 10 | 580 / 1,769 / 5,847 |

**Computing environment.** We implemented all models using PyTorch (Paszke et al., 2019), Py-Torch Geometric (Fey & Lenssen, 2019), and Scikit-learn (Pedregosa et al., 2011). All data sets used in the experiments are obtained from PyTorch Geometric. We conduct the experiments on a server with four NVIDIA RTX A6000 GPUs (48GB memory each).

**Implementation details.** For fair comparison, we run all models on the entire data set under the transductive semi-supervised setting. All models are initialized with Glorot initialization (Glorot & Bengio, 2010) and are trained using the Adam optimizer (Kingma & Ba, 2015). For reporting the silhouette coefficient, k-means is run for 1000 epochs. For all models on all data sets, the $\ell_2$ weight decay factor is set to $5 \times 10^{-4}$ and the number of training epochs is set to 400. For all models, we use the early stopping strategy on the F1 score on validation set. In all experiments, we use $\widehat{\mathbf{A}} = (\mathbf{D} + \mathbf{I})^{-1}(\mathbf{A} + \mathbf{I})$, where $\mathbf{D} = \mathrm{diag}(\sum_j \mathbf{A}_{ij})$. For FlowGMM, GC-Flow, and its variants, we clip the norm of the gradients to with the range $[-50, 50]$. For GC-Flow-l and GC-Flow-p, since $\widehat{\mathbf{A}}^{(j)}$ in each flow may not has a full rank, we add to $\widehat{\mathbf{A}}^{(j)}$ a diagonal matrix with damping value $10^{-3}$. Moreover, the slope in $\mathrm{LeakyReLU}$ is set to $0.2$. In all models involving normalizing flows, we use RealNVP (Dinh et al., 2017) with coupling layers implemented by using MLPs. Following Izmailov et al. (2020), the mean vectors are parameterized as some scalar multiple of the vector of all ones, and the covariance matrices are parameterized as some scalar multiple of the identity matrix. On the other hand, the GMM models are implemented by using Scikit-learn with full covariance matrices. The number of training epochs for GMMs is set to 200.

Too large a feature dimension renders a challenge on the training of a normalizing flow. Hence, we perform dimension reduction in such a case. For Cora, Pubmed, Computers, and Photo, we use PCA to reduce the feature dimension to 50; and for Citeseer, to 100. We keep the dimension 300 on Wiki-CS without feature reduction.

**Hyperparameters.** We use grid search to tune the hyperparameters of FlowGMM, GC-Flow, and its variants. The search spaces are listed in the following:

- Number of flow layers: 2, 4, 6, 8, 10, 12, 14, 16, 18, 20;
- Number of dense layers in each flow: 6, 10, 14;
- Hidden size of flow layers: 128, 256, 512, 1024;
- Weighting parameter $\lambda$: 0.01, 0.05, 0.1, 0.15, 0.2, 0.25, 0.3, 0.35, 0.4, 0.45, 0.5;
- Gaussian mean and covariance scale: $[0.5, 10]$;
- Initial learning rate: 0.001, 0.002, 0.003, 0.005;
- Dropout rate: 0.1, 0.2, 0.3, 0.4, 0.5, 0.6.

Additionally, for GC-Flow-p and GC-Flow-l:

- Number of dense layers in $\mathbf{E}_1$ and $\mathbf{E}_2$: 4, 6;
- Hidden size of dense layers in $\mathbf{E}_1$ and $\mathbf{E}_2$: 128, 256;
- Embedding dimension $d$: 8, 16.

For GCN and GraphSAGE, we set the hidden size to 128, dropout rate to 0.5, and learning rate to 0.01. For GAT, we follow Veličković et al. (2018) to set the hidden size to 8, the number of attention heads to 8, droupout rate to 0.6, and the learning rate to 0.005.

The results in Table 1 for GC-Flow are obtained by using different number of flows and number of layers per flow for different data sets. For Computers, Photo, and Wiki-CS, we use 4, 2, and 2 flows, respectively, with 6 dense layers in each flow. For Cora, Citeseer and Pubmed, we use 4, 10 and 10 flows, respectively, with 10 dense layers in each flow.

The results in Table 3 also use different number of flows and layers. For GC-Flow-p, on Cora, Citeseer, Pubmed, Photo, and Wiki-CS, we use 10 flows; while on Computers, we use 6 flows. For GC-Flow-1, we use 2 flows for Wiki-CS and 4 flows for the rest of the data sets. For both For GC-Flow-p and GC-Flow-l, there are 6 dense layers per flow on Wiki-CS while 10 per flow on the rest of the data sets.

# F ADDITIONAL EXPERIMENT RESULTS

**Clustering performance.** Table 5 compares the clustering performance between GC-Flow and various contrastive-based GNN methods, for Citeseer and Pubmed. GC-Flow maintains the attractively best performance on the silhouette score, which measures cluster separation. For the cluster assignment metrics, GC-Flow remains competitive on ARI, which measures cluster similarity, while falling back on NMI, which measures cluster agreement.

Table 5: Clustering performance of various GNN methods. The two best cases are boldfaced. Data sets: Citeseer (top) and Pubmed (bottom).

|  | NMI | ARI | Silhouette |
|---|---|---|---|
| DGI | $\mathbf{0.427 \pm 0.001}$ | $0.399 \pm 0.002$ | $\mathbf{0.314 \pm 0.000}$ |
| GRACE | $0.380 \pm 0.017$ | $0.378 \pm 0.023$ | $0.219 \pm 0.014$ |
| GCA | $0.336 \pm 0.008$ | $0.279 \pm 0.005$ | $0.251 \pm 0.015$ |
| GraphCL | $0.417 \pm 0.001$ | $0.372 \pm 0.002$ | $0.299 \pm 0.001$ |
| MVGRL | $\mathbf{0.468 \pm 0.002}$ | $\mathbf{0.445 \pm 0.004}$ | $0.305 \pm 0.000$ |
| GC-Flow | $0.405 \pm 0.013$ | $\mathbf{0.426 \pm 0.014}$ | $\mathbf{0.538 \pm 0.022}$ |

|  | NMI | ARI | Silhouette |
|---|---|---|---|
| DGI | $0.379 \pm 0.001$ | $0.361 \pm 0.002$ | $0.426 \pm 0.001$ |
| GRACE | $0.305 \pm 0.075$ | $0.261 \pm 0.111$ | $0.149 \pm 0.008$ |
| GCA | $\mathbf{0.488 \pm 0.016}$ | $\mathbf{0.511 \pm 0.030}$ | $0.354 \pm 0.014$ |
| GraphCL | $0.337 \pm 0.001$ | $0.308 \pm 0.002$ | $0.412 \pm 0.002$ |
| MVGRL | $\mathbf{0.391 \pm 0.001}$ | $0.361 \pm 0.001$ | $\mathbf{0.441 \pm 0.001}$ |
| GC-Flow | $0.384 \pm 0.011$ | $\mathbf{0.460 \pm 0.015}$ | $\mathbf{0.655 \pm 0.013}$ |

**Training behavior.** Figure 5 plots the convergence of the training loss for various methods. The loss for GCN is the cross-entropy while the loss for FlowGMM and GC-Flow is the likelihood. All the curves are smooth and exhibit reasonable decay, suggesting convergence as expected. GC-Flow converges similarly to FlowGMM, while GCN converges faster.

**Visualization of the representation space.** Figure 6 shows the t-SNE plots for data sets Citeseer, Computers, Photo, and Wiki-CS, one per row.

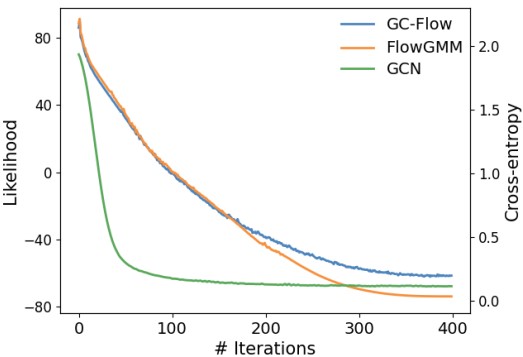

Figure 5: Convergence of the training loss (Cora).

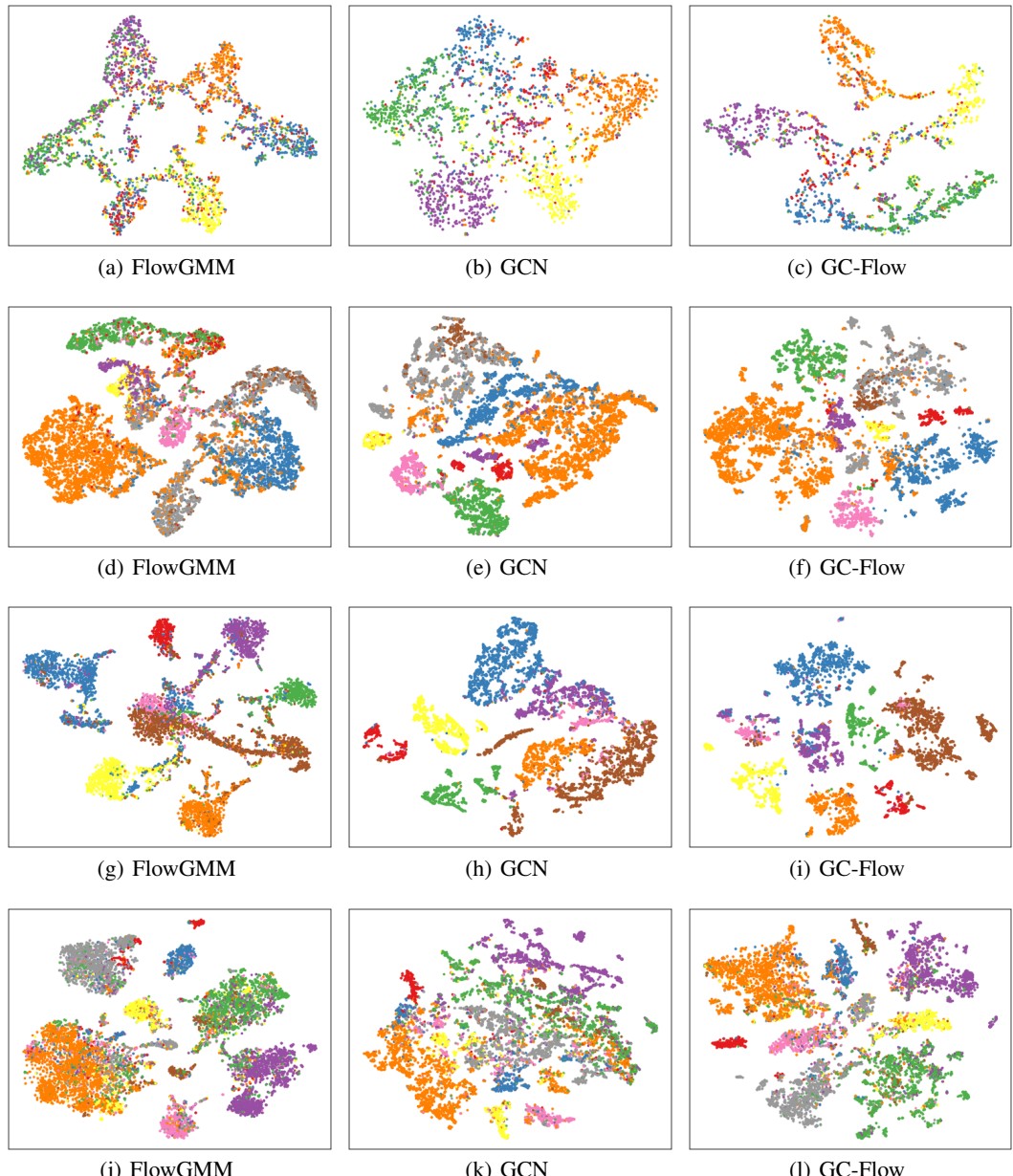

Figure 6: Representation space of several data sets under different models. Data set from top to bottom: Citeseer, Computers, Photo, and Wiki-CS.

