# OpenReview forum: "Graph Convolutional Normalizing Flows for Semi-Supervised Classification and Clustering"
_ICLR.cc/2023/Conference — Submitted to ICLR 2023_

### Official Review · Reviewer_qGKA · 2022-10-18

**Confidence:** 4
**Correctness:** 4
**Technical Novelty And Significance:** 4
**Empirical Novelty And Significance:** 3
**Recommendation:** 8

**Clarity, Quality, Novelty And Reproducibility:**

Clarity:  The paper is well-organized and clearly written.

Quality: The paper is technically sound; the important derivations and detailed variants are provided in the appendix.

Novelty: This work provides some interesting insights into the node representation of graphs from the view of clustering. The approach is novel.

Reproducibility: The empirical results seem reasonable, and I don’t doubt the reproducibility.


**Strength And Weaknesses:**

Strengths:
1. This work is quite interesting. The paper provides a new angle to understand the graph node representation from clustering by the nature of normalizing flows, which is rarely considered in GNNs.
2. The proposed GC-Flow is simple and practical on various datasets. In addition, two variants are studied which show the flexibility of the GC-Flow framework.
3. Extensive experiments show the effectiveness of the proposed method. Some of the experimental results are inspiring. The hyperparameter tuning in the appendix is clear and reasonable.
4. The appendix contains solid derivations for Lemma 1, which is well organized and clear to understand.

Weaknesses:
1. In Eq.(8) and Eq.(9), the authors define the class-conditional likelihood $p(x_i|y_i=k)$ and marginal likelihood $p(x_i)$. However, the total loss function is not illustrated clearly. It is better to briefly describe a concrete formulation of the loss function for GC-Flow.
2. There is no analysis of complexity. Scalability is important for flow-based generative models. As the paper described, GC-Flow is based on normalizing flows, and there is a need to calculate the Jacobian determinant. Considering the #train/val/test listed in Table 3, I am wondering if only a part of the samples for each data set were used for the empirical study. I suggest the authors add complexity analysis to improve this paper.
3. Some important related works are missing. It is better for the authors to discuss some differences and connections between GC-Flow and the method [1] which also aims to model distributions of graph-structured data using normalizing flows. Besides, related works on node clustering [2] are missing.

[1] Deng, Z., Nawhal, M., Meng, L., & Mori, G. (2019). Continuous graph flow. arXiv preprint arXiv:1908.02436.

[2] Fettal, C., Labiod, L., & Nadif, M. (2022). Efficient Graph Convolution for Joint Node Representation Learning and Clustering. WSDM.


**Summary Of The Paper:**

This paper presents GC-Flow, a generative framework to generate the representation of graphs. The framework is effective for node classification and unravels the inherent structure of data for clustering. More specifically, GC-Flow takes both the advantages of normalizing flows and graph convolutions. For prediction, it computes the class conditional likelihood p(x|y) and applies the Bayes rule together with the class prior p(y). Experimental results have shown the effectiveness of various benchmarks compared with the related baselines.

**Summary Of The Review:**

The paper is strong, and I recommend it for acceptance. The proposed method is simple and practical. It also provides interesting insights into normalizing flows and GNNs. However, there is still room for improvement. I kindly hope the authors consider the points highlighted in my reviews.

---

> ### Author Response · Authors · 2022-11-16
> **Response to Reviewer qGKA**
>
> Thank you for the feedback! We have updated the paper to address your comments. Please see the separate post “[Updates of the paper and significance statements](https://openreview.net/forum?id=3i9EgUss-Vs&noteId=26Db12BNHk)” for a summary of the changes. We respond to your individual questions below.
>
> **RE: The total loss function.**
>
> The total loss function appears is Eqn (5). Later, Eqn (8) and Eqn (9) derive the labeled and unlabeled terms used in Eqn (5), respectively. We have edited the paper to remind the reader of the loss function.
>
> **RE: Complexity analysis.**
>
> Thank you for the suggestion. We have now conducted a complexity analysis and included it in Appendix D. The computational cost of the matrix determinant is very lightweight; it is not $O(D^3)$ but only $O(D)$, because the Jacobian matrix is triangular. Instead, the cost of the Jacobian determinant is dominated by the evaluation of the neural network that parameterizes the normalizing flow (e.g., an MLP). Overall, the computational cost of the proposed GC-Flow is comparable to that of GCN; these two costs admit the same scaling behavior.
>
> **RE: Question “I am wondering if only a part of the samples for each data set were used for the empirical study.”**
>
> For all methods, we conduct full-batch training. The training objective for GC-Flow is a likelihood loss and we use all the nodes in the graph (either labeled or not) to compute this loss. In the updated paper, we have reorganized Section 4.3 into two sections in the appendix (C and D), due to the expansion of materials and the limitation on space.
>
> **RE: Related works.**
>
> We have included a discussion of several papers suggested by you and other reviewers in the Related Work section. Unlike ours, the work of Deng et al. (2019) is a continuous flow that admits different computational complexities and dynamics from the discrete flows that we are concerned with in this paper (essentially the cost comparison is the trace of a free-form Jacobian versus the determinant of a structured Jacobian). The clustering paper, together with other contrastive-based GNN papers, is also discussed.

---

> > ### Comment · Reviewer_qGKA · 2022-11-23
> > **Comments and Final Decisions**
> >
> > Thanks to the author for such detailed and elaborate responses to my queries. Sections C and D addressed my concerns. I believe the paper has improved a lot.
> >
> > I agree with the significance stressed by the authors. In the semi-supervised setting, the dominant GNNs are discriminative models weak in clustering. In contrast, GC-Flow, as a type of GNN, is a generative model that learns the embedding Z from the data distribution p(x,y), and unveils well-separated clusters in the embedding. Overall, I think this work is innovative.
> >
> > In addition, the results of additional experiments provided in the updated manuscript are noticeable. Comparisons with the leading contrastive methods show its superiority in clustering. Considering all information, I still stand for acceptance of this work.

---

### Official Review · Reviewer_kEff · 2022-10-21

**Confidence:** 4
**Correctness:** 4
**Technical Novelty And Significance:** 2
**Empirical Novelty And Significance:** 2
**Recommendation:** 5

**Clarity, Quality, Novelty And Reproducibility:**

This paper is clearly written and easy to follow, it has a clear step-by-step derivation of its normalizing flow model. The method it proposes is innovative in the graph domain. However, as I mentioned before, this paper lacks a clear definition of what kind of problem it wants to solve, if it only focuses on improving the clustering performance of GNN under semi-supervised learning, I do not think it is innovative enough.

**Strength And Weaknesses:**

Strengths:
1. The proposed GC-Flow model is an innovative generative model on the graph data, which considers the interdependency among instances (nodes) and incorporates it into the normalizing flow model.
2. The improvement in the clustering results is very significant, and the visualization demonstrates that GC-Flow has a more interpretable latent representation.

Weakness:
1. The main drawback of this paper lies in that it fails to identify a significant problem in the current GNN models. I admit the method proposed is interesting, but I cannot see the significance of the problem it studies. If the main purpose of adding normalizing flow is to increase the clustering performance, why do not consider adding the community partition losses/clustering losses to the supervised losses? Adding normalizing flow is not that straightforward to realize this goal.

   As pointed out by FlowGMM, it does not actually increase the performance too much but treats interpretability and broad applicability as its main advantage. I think it is unnecessary to introduce normalizing flow on GNN to improve the clustering performance only.

2. The baselines are too weak and graph contrastive learning methods [1,2,3] are totally missing from the baselines, which are expected to have good performance in both clustering and node classification. Both [2] and [3] have the loss to explicitly encourage clustering.

[1] Graph Contrastive Learning with Adaptive Augmentation. Zhu et al., WWW 2021.

[2] Graph Communal Contrastive Learning. Li et al.,  WWW 2022.

[3] X-GOAL: Multiplex Heterogeneous Graph Prototypical Contrastive Learning, Jing et al., CIKM 2022.

**Summary Of The Paper:**

This paper focused on learning a structured node representation on top of a GNN model. It follows the idea of FlowGMM and applies normalizing flow to the GNN model. The proposed GC-Flow model leads to significantly better clustering results and improves node classification.

**Summary Of The Review:**

This paper has good writing and is well-organized. The model proposed is very interesting to the community and demonstrates strong power in improving the clustering performance of current GNN models. But generally, I feel the authors do not have a clear definition of what kind of significant problem they want to solve, or they do not explore the benefits of their model enough. The baselines in the experiment are also too weak. I vote for the rejection by this time unless the authors could clearly tackle the weakness I mentioned before.

---

> ### Author Response · Authors · 2022-11-16
> **Response to Reviewer kEff**
>
> Thank you for the feedback! We have updated the paper to address your comments. Please find the response to your individual questions below.
>
> **RE: What is a significant problem in the current GNN models?**
>
> Please see the separate post “[Updates of the paper and significance statements](https://openreview.net/forum?id=3i9EgUss-Vs&noteId=26Db12BNHk)” for discussions on why we believe this work contributes knowledge to the field and advances semi-supervised learning.
>
> **RE: Why do not consider adding the community partition losses/clustering losses to the supervised losses?**
>
> We believe that different models and training techniques have their pros and cons. Adding a loss that encourages clustering is a useful approach to improving clustering, but it is not a unique approach. The clustering loss may need to be balanced with other objectives exhibited in the other loss terms through tuned weighting. The loss mechanism may be considered as an external forcing to achieve multiple objectives, clustering being one.
>
> Our model, on the other hand, improves clustering through internal modeling. It comes from a different angle, which addresses the distinction between discriminative models versus generative models, the latter of which has a solid Bayesian ground but is less explored by the GNN literature. The Gaussian mixture is a very natural model for clustering and a normalizing flow enhances its power through parameterized transformations. As opposed to the external forcing mechanism imposed by the training loss, our approach unveils the internal structure of the data by modeling the cluster conditional likelihood p(x|y) and the cluster prior p(y), which a discriminative GNN model lacks.
>
> **RE: Benefits of the proposed model.**
>
> Being generative, our model returns more information on the clustering structure of data, including the likelihood p(x|y) that a node belongs to a particular cluster and the prior distribution p(y) of the clusters. This is more informative than other graph clustering approaches (including contrastive methods) that return hard/soft cluster assignments. This information increases the interpretability of the internal structure of the data.
>
> **RE: Graph contrastive learning baselines.**
>
> Thank you for the suggestions. We have included an additional experiment to compare the proposed model with contrastive-based methods (see Table 2 and Table 5). We are unable to locate codes for two of the methods you suggested; instead, we collect five methods, most of which are published at the recent top conferences, for experiments. The results suggest that our model remains superior under the silhouette metric (which measures cluster separation) and is competitive under the ARI (which measures cluster similarity) and NMI (which measures cluster agreement) metrics. Our method is particularly performant on the structure evaluation, owing to the use of a mixture model for the representation space.
>
> Additionally, we have included a discussion of the contrastive papers you mentioned in the Related Work section.

---

> > ### Comment · Reviewer_kEff · 2022-11-20
> > **Thanks for the response**
> >
> > I want to thank the reviewers for the detailed response to my concerns. However, my concern still remains.
> >
> > First, I do not understand the terms 'external modeling' and 'internal modeling' since normalizing flow also relies on the loss to encourage the clustering, the loss of normalizing flow is just minimizing the distance between the manifold distribution and mixture of Gaussian, there shall be no difference in essence here.
> >
> > Second, even from the generative modeling part, GC-Flow is also unnatural. The community partition loss makes sense since we have the fundamental assumption that the nodes from the same community tend to have a strong connection to each other, which leads to the clustering in the graph. In terms of generation, each time we have a new node, we can have its link probability to other nodes based on their community attributes. This is a process of 'node to edge': we first have a user account in the social network, then it will get connected to others given its own attribute.
> >
> > While for GC-Flow, the adjacency matrix is fixed, which means the cluster assignment for each node may have been decided at the beginning of the generation, what normalizing flow could do is to make the node features from the same cluster as close as possible. This is like a process of 'edge to node features', which is very unnatural in the graph generation process. So I think the generation side of normalizing flow is in fact very weak and how interpretable it could be beyond the T-SNE visualization is in question.

---

> > > ### Author Response · Authors · 2022-11-21
> > > **Response to further concerns**
> > >
> > > Dear reviewer, thank you for the follow-up. It might be helpful to disambiguate your concerns from the angles of discriminative models versus generative models, as your questions are more general than those specifically focusing on graphs.
> > >
> > > The fundamental difference between them is that discriminative models learn the boundary between classes, whereas generative models learn the data distribution. The wording "generative" describes how existing data are generated, not necessarily how one generates new data (although the recent literature tends to gear toward the latter more) [1]. See also this [Wikipedia page](https://en.wikipedia.org/wiki/Generative_model#Contrast_with_discriminative_classifiers). If one really cares about new generation, for a citation network for example, the relevant capability of our model is to generate new papers that follow a given citation structure. Such a capability is a bit artificial and we do not aim at it in this work. Rather, our aim is to obtain a probabilistic model that explicitly describes the distribution of the known data. The interpretability comes from the unraveling of this distribution, which would not be provided by other mechanisms such as contrastive losses or partition losses. Hope this would answer your second concern about data generation.
> > >
> > >
> > > Regarding your first concern, it would be worthwhile to elaborate on the data distribution $p(x,y) = p(x|y)p(y)$. The wording “internal” refers to the modeling of the distribution itself. In other words, clustering is only a byproduct of the innate distribution of the data, as one can apply the Bayes rule $p(y|x) \propto p(x|y)p(y)$ after the distribution is learned. On the other hand, discriminative models do not learn the distribution. All it does is to use some loss function to orient a better agreement between $p(y|x)$ and the labels. Such does not reveal the data distribution and hence we call the training “external”.
> > >
> > > We would also like to clarify that your statement “the loss of normalizing flow is just minimizing the distance between the manifold distribution and mixture of Gaussian” is subtly mistaken. The Gaussian mixture is in the representation space (variable z), while the distribution we learn is in the data space (variable x). The normalizing flow is a vehicle to compute the density of x through the density of z (and vice versa). Optimizing the loss by no means is equivalent to minimizing the distance between the manifold distribution and the Gaussian mixture. Rather, it is to maximize the likelihood of observing data (either labeled (x,y) or unlabeled x), given an assumed structure of the representation space z (Gaussian mixtures).
> > >
> > > Additionally, we have included comparisons with five contrastive methods and show our empirical advantage. The comparatively high silhouette score that measures cluster separation is compelling. We believe that our simple yet effective method contributes positively to the knowledge of the field regarding how generative models can be built for graph representation learning.
> > >
> > > [1] Liu, Jenny, et al. "Graph normalizing flows." NeurIPS 2019.

---

### Official Review · Reviewer_bfhQ · 2022-10-24

**Confidence:** 3
**Clarity, Quality, Novelty And Reproducibility:** please see the strength and weakness.
**Correctness:** 3
**Technical Novelty And Significance:** 2
**Empirical Novelty And Significance:** 2
**Recommendation:** 5

**Strength And Weaknesses:**

Strength:
1)	The paper is well organized.
2)	The idea of combining graph convolution into normalizing flow and using Gaussian mixture distribution to encourage clustered representation learning is interesting,
3)	Experimental results demonstrate that the representations learned under the proposed method appear to be more clustered than the previous GNN models, in addition to maintaining classification performance.


Weaknesses
1)	The proposed method can be viewed as a direct combination of GCN and normalizing flow, with the ultimate transformed distribution, which is Gaussian in conventional NF, replaced by Gaussian mixture distribution, encouraging the latent representation to be more clustered. Technically, there is no enough new stuffs here.
2)	More Seriously, to ensure the intractability of the normalizing flow after absorbing the graph neural network, the proposed model has to replace the basic operation \sigma(AXW) with the operation \sigma(AX) in the graph neural networks, abandon the feature affine transformation operation, i.e., XW, before passing the intermediate representations to neighboring nodes. Since the W is the main parameters to be learned in GNN, abandoning it means the representation ability of GNN is restricted significantly. The experimental results also show that the proposed model brings very little gains over the old models like GCN and GAT on classification tasks.
3)	Without using the feature affine transformation AXW, then the dimension of intermediate hidden representations will always be kept the same as that of input feature since the NF have to maintain the dimension unchanged. Then, if the dimension of input feature is very high, in addition to the complexity issue, the learned feature will be also be very high, which may not be very useful as nowadays we often expect the learned features to be compact.
4)	For the experiments, since the paper want to demonstrate the proposed model is able to learn clustering-friendly representations, we expect to directly see how the model performs on clustering task on the clustering performance metric, like accuracy ACC, normalized mutual information NMI etc, rather than the indirect Silhouette criteria, which is not meaningful at all.


**Summary Of The Paper:**

This paper proposed to use the combination graph convolution and normalizing flows to replace the traditional GNN layers. Comparing with the conventional normalizing flow that transform an original x into a latent z (usually Gaussian distributed), there are two main differences here. First, instead of only transforming the original data x, at each intermediate step of transformation, the model will first fuse the data from neighboring nodes and then transform the fused data using a standard NF transformation. Second, different from conventional NF that intend to eventually transform the raw data x into a Gaussian distributed representation, the proposed method transforms data into mixture Gaussian distribution, hoping to produce representations that are friendly for clustering. A set of experiments were conducted to show the proposed model could be simultaneously good at classification and clustering.

**Summary Of The Review:**

The proposed method looks like a direct combination of GNN and NF, making the technical contribution here limited. To guarantee the tractability of NF, the proposed model has to abandon the transformation of feature operation XW before passing to neighbors, making the modeling ability of GNN reduced significantly. The experiments cannot support the claims or objectives, neither.

---

> ### Author Response · Authors · 2022-11-16
> **Response to Reviewer bfhQ**
>
> Thank you for the feedback! We have updated the paper to address your comments. Please find the response to your individual questions below.
>
> **RE: What is new.**
>
> Please see the separate post “[Updates of the paper and significance statements](https://openreview.net/forum?id=3i9EgUss-Vs&noteId=26Db12BNHk)” for discussions on why we believe this work contributes knowledge to the field and advances semi-supervised learning.
>
> **RE: Comment “abandoning the feature affine transformation operation means the representation ability of GNN is restricted significantly.”**
>
> The representation ability of our model is not expected to be restricted, because the feature affine transformation in GNNs is replaced by a normalizing flow, an invertible function that by itself is parameterized. Just like feed-forward neural networks, which contain several layers of affine transformations, are universal approximators, a normalizing flow can also universally approximate functions. A difference is that an affine transformation can change the dimension of the spaces, but normalizing flows cannot. In exchange, the invertibility of normalizing flows allows computing data density exactly, while affine transformations generally cannot.
>
> **RE: Comment “if the dimension of input feature is very high, in addition to the complexity issue, the learned feature will be also be very high, which may not be very useful as nowadays we often expect the learned features to be compact.”**
>
> It is indeed the case that many normalizing flows (for tabular data) suffer high dimensionality; however, this is not a practical drawback for two reasons. First, one may reduce the input dimensionality through preprocessing (such as PCA), as is what we did in the experiments. This would be needed for some standard benchmark datasets (such as CORA), where the features are a bag of words. As the features switch to more realistic cases, such as fixed-dimensional word embeddings, the dimensions can reasonably be handled by normalizing flows and no dimension reduction is needed.
>
> Second, in many cases, the learned features are not the aim but a means. If the aim is clustering, high-dimensional spaces pose more challenges for metrics that measure cluster separation, because distances are concentrated [1]. In this case, our outstanding silhouette coefficient over GNNs and contrastive methods demonstrates that the model works really well in forming clusters (for example, in Table 1, our model doubles the silhouette score obtained by GNNs for Cora and triples for Wiki-CS).
>
> [1] Aggarwal et al. On the Surprising Behavior of Distance Metrics in High Dimensional Space. LNCS 1973, pp. 420--434, 2001.
>
> **RE: Evaluation metrics.**
>
> There may be some misconceptions about the evaluation metrics for clustering. The silhouette coefficient is a standard metric for evaluating clustering when no ground truths are known. It measures cluster separation. We use this metric mainly to complement the case when ground truths are known. That said, in the revision, we have included an additional experiment (Table 2 and Table 5) and compared with five recent graph-based clustering methods, wherein we used NMI and ARI metrics (requiring ground truths), in addition to the silhouette. Our method is particularly performant on cluster separation, owing to the use of a mixture model for the representation space.

---

### Official Review · Reviewer_cUhi · 2022-10-25

**Confidence:** 3
**Clarity, Quality, Novelty And Reproducibility:** The author provide a quite complete s…
**Correctness:** 3
**Technical Novelty And Significance:** 3
**Empirical Novelty And Significance:** 3
**Recommendation:** 5

**Strength And Weaknesses:**

Strength:
The paper proposes a coherent story, etc

Weakness:
1 replace GNN layer with graph convolutions & normalizing flows will significantly add more computational burden? especially inverse of Jacobians? Will the normalizing flow harder to train due to  intermediate stage of graph functions' ill behaviour (due to graph convolution), leading to instability and divergence of the algorithm?
Have the author consider other priors than gaussian mixture?


2. The paper reports microF1 score in Table 1

How does the paper’s performance compare with leaderboard here
https://paperswithcode.com/sota/node-classification-on-pubmed
for example, pubmed acc is 91+

how does performance compare with https://arxiv.org/pdf/2109.05641v1.pdf
https://arxiv.org/pdf/2012.06113.pdf (Table 3)
seems much better than Table 1 of the paper.
Can the author clarify?




**Summary Of The Paper:**

The paper proposes a method to replace a GNN layer by a combination of graph convolutions and normalizing flows under a Gaussian mixture representation space. The proposed method not only classifies well, but also yields node representations that capture the inherent structure of data, as a result forming high-quality clusters (i.e. cluster well).


**Summary Of The Review:**

I am not fully on board on the motivation of this and the results section.

---

> ### Author Response · Authors · 2022-11-16
> **Response to Reviewer cUhi**
>
> Thank you for the feedback! We have updated the paper to address your comments. Please find the response to your individual questions below.
>
> **RE: Motivation.**
>
> Please see the separate post “[Updates of the paper and significance statements](https://openreview.net/forum?id=3i9EgUss-Vs&noteId=26Db12BNHk)” for discussions on why we believe this work contributes knowledge to the field and advances semi-supervised learning.
>
> **RE: Question “replace GNN layer with graph convolutions & normalizing flows will significantly add more computational burden? especially inverse of Jacobians?”**
>
> Overall, the cost of evaluating the loss of the proposed model is comparable to that of GCN; and these two costs admit the same scaling behavior. We have included a complexity analysis in Appendix D. One common misconception is that normalizing flows involve the computation of the Jacobian and they are forbiddingly costly. We clarify that the popularly used normalizing flows all impose a structure on the Jacobian (typically triangular), such that computing the determinant is not $O(D^3)$ but only $O(D)$ (where $D$ is the feature dimension). Also note that the Jacobian does not need to be inverted, because the determinant of the inverse Jacobian is equal to the inverse of the determinant of the Jacobian. In fact, the dominant cost of the Jacobian is instead the evaluation of the neural network that parameterizes the flow (for example, an MLP). Such an MLP often has a small number of layers (say, 3) and its cost is quite comparable to a dense feature transformation in GCN.
>
> **RE: Question “Will the normalizing flow harder to train due to intermediate stage of graph functions' ill behaviour (due to graph convolution), leading to instability and divergence of the algorithm?”**
>
> GC-Flow is not hard to train. In the updated paper (Appendix F), we have included an experiment to show the typical convergence behaviors of GC-Flow and GCN. Both of them admit a nice loss curve.
>
> **RE: Question “Have the author consider other priors than gaussian mixture?”**
>
> We do not consider other priors, because the Gaussian mixture is the most natural choice for clustering---each Gaussian corresponds to one cluster. That said, in principle, one may use a mixture of any distributions, if some prior knowledge of the shape of each cluster exists. In this case, one needs to replace the Gaussian density in the probability model (Eqn (8) and (9)) by the density of the corresponding distribution.
>
> **RE: Performance.**
>
> Compared with classification, our model shows stronger advantages in clustering, because the mixture prior is used to encourage a good clustering structure---tighter clusters and more separations among clusters. Our model can match the classification performance of the early representative GNNs (GCN, GraphSAGE, GAT, etc), but as the literature keeps pushing the classification accuracy, it is not surprising that many current models can outperform ours (including the methods you mentioned).
>
> However, the clustering performance of our model is outstanding under the metric of silhouette coefficients. In the updated paper (Table 2 and Table 5), we further included additional experiments to compare with five contrastive-based methods, which had been shown to produce rather competitive clustering results. Our model still outperforms them substantially on cluster separation (silhouette), while being competitive on cluster similarity (ARI) and cluster agreement (NMI).
>
> It is possible to improve the classification performance of our model through more sophisticated graph convolutions, such as the adaptive channel mixing you mentioned, where the architecture is separated into a low-pass filter component and a high-pass filter component. Every such extension may need a separate calculation of the Jacobian determinant, for which our work serves as a framework and provides an example.

---

### Author Response · Authors · 2022-11-16
**Updates of the paper and significance statements**

**To all reviewers:**

Thank you for your comments in the initial round of review. We are encouraged that you find the normalizing flow idea interesting and novel, the exposition is well organized, and the clustering results are significant. We will separately respond to the questions under your review posts. Here is a summary of the major updates of the paper based on your comments.

- Edited the Introduction section to highlight the significance of the work
- Added complexity analysis, which suggests a similar training cost compared with GNNs
- Added related work and discussions
- Added an experiment to illustrate the training behavior
- Added an experiment to compare with graph contrastive learning methods for clustering. Additional metrics were used (NMI and ARI, requiring ground truths), besides the silhouette coefficient (not requiring ground truths).

**Significance statements.** Additionally, we stress a few points regarding the motivation and the significance of this work here, as some of you are concerned about.

First, there are two classes of predictive models: discriminative and generative. In graph-based deep learning (which we use as a synonym for semi-supervised learning), discriminative models are dominant, as one only needs to use a graph neural network plus a softmax activation to model p(y|x) and to predict. This work is an effort to complement the literature with a generative model that specifies not only how classes/clusters are predicted, but also the distribution of the data, p(x,y). Such a model increases the understanding of the data and makes predictions more interpretable.

Second, while classification problems prevail on graph data, clustering is a weakness for many of the standard graph neural networks. One reason is that the cross-entropy loss for classification is not effective enough for clustering. Indeed, many papers that study clustering propose using a clustering loss, a contrastive loss, or even a combination of the two. Rather than tackling the problem through the training loss, which is external to the network architecture, we directly propose a neural network (normalizing flow) as an internal solution. The neural model transforms the probability density of the observed data to match that of the clusters underlying the data, with density being exactly computed, and possible to optimize, for example via maximum likelihood.  A reviewer is confused that in our model, the weight matrix W disappears from the GCN, which seems to weaken the expressiveness. However, each constituent flow by itself is a neural network, whose parameters make up the disappearance of W in GCN; it is equally powerful in terms of function approximations. There is no mystery why some graph neural networks (such as GCN) work less well for clustering but some (such as ours) work well: it is the training objective that matters.

Third, normalizing flows are a very useful generative model. In the literature, we see that most of the papers on the intersection of normalizing flows and graphs aim at generating new graphs; fewer papers address node-level tasks. The most relevant papers to ours are GNF (Liu et al., 2019) and CGF (Deng et al., 2019). As discussed in the Related Work section, the former uses the adjacency matrix A to parameterize the st-networks in a coupling flow (while we use A to transform the layer input), and the latter is a continuous flow that is computationally more expensive than a discrete flow. On the other hand, our model is very simple (yet effective), with elegant math on the calculation of the Jacobian determinant. We believe it contributes to the knowledge of the field and is a practical tool for applications.

Fourth, clustering can be measured by using many metrics, each of which focuses on a different aspect. The general literature uses NMI and other metrics that measure cluster assignment, through the use of ground truth labels. We would stress that such metrics do not measure the clustering structure (e.g., separation of clusters), for which the silhouette coefficient is more appropriate. Our model improves the silhouette coefficient significantly over not only standard GNNs, but also those trained with contrastive losses.

---

### Decision · Program_Chairs · 2023-01-20

**Decision:**

Reject

**Justification For Why Not Higher Score:**

Three out of four reviewers recommend rejection and I do not have the grounds to go against their recommendation.

**Justification For Why Not Lower Score:**

N/A

**Metareview: Summary, Strengths And Weaknesses:**

Summary:
This paper combines graph convolutions and normalizing flows as a replacement of GNN layers, with a representation space where the latent variable follows a Gaussian mixture distribution, making the representation suitable to represent clusters.

Strengths:
The paper is well written, and the combination of flow and graph convolution interesting. The experimental results for clustering are encouraging.

Weaknesses:
The reviewers found the technical novelty of the paper to be limited, while the experimental results were not quite convincing. In part, important baselines are missing, but more importantly, the main issue is that the paper does not seem to solve a clear and significant problem in the GNN literature.